# Systematic identification of variant-specific RNA structure-small molecule interactions exemplified by RNA G-quadruplexes

Emi Miyashita [1,2], Kazumitsu Onizuka [3] ✉, Yutong Chen[3], Hiroki Yoshida [2], Hina Hatayama[3], Shunya Ishikawa[3], Peijie Yan[2,4], Takahito Hasegawa[3], Mamiko Ozawa[3], Kaho Maeta [2], Fumi Nagatsugi [3], Hirohide Saito [1,4] ✉ & Kaoru R. Komatsu [2] ✉

Individual genetic variations, such as cancer-associated somatic mutations, alter RNA structures, thereby potentially enhancing or inhibiting the binding of RNA-targeting small molecules. However, to date, no approach has been available to identify these variant-specific RNA-small molecule interactions due to technical limitations. Here, we present Binding- and Vinyl-Quinazolinone-Induced Deletion-Based Mutational Profiling (BIVID-MaP), a high-throughput method for detecting RNA-small molecule interactions that combines binding-dependent covalent modification with profiling of deletions upon reverse transcription via deep sequencing. Using BIVID-MaP, we uncovered numerous variant-specific interactions between a G-quadruplex (G4)-binding small molecule and RNAs harboring single-nucleotide variants. Several cancer-associated somatic mutations significantly influence the binding intensity of a small molecule by affecting target G4 structures. These results demonstrate that BIVID-MaP can reveal previously ignored variant-specific RNA-small molecule interactions affected only by a single-nucleotide mutation, which may contribute to the development of RNA-targeting drugs in the future.

RNA-binding small-molecule drugs bind to structured regions of RNA and offer a promising therapeutic strategy for regulating genes not easily modulated by traditional protein-targeting drugs[1]. Developing these RNA-targeting compounds requires rigorous evaluation of target specificity and selectivity, thus underlining the need for comprehensive methods to profile RNA-small molecule interactions[2,3].

Notably, single-nucleotide variants (SNVs) and somatic mutations can significantly influence the formation of these structured RNA regions[4,5]. Such variant-induced structural alterations can affect the interaction of small-molecule drugs, potentially impacting their therapeutic efficacy. Recent observations revealed that branaplam,

discontinued from clinical trials for Huntington's disease, exhibits distinct splicing-modulating profiles in patient-derived cells carrying SNVs[6]. This finding highlights the importance of evaluating both on-target and off-target effects mediated by genetic variations during the development of RNA-targeting drugs. Therefore, advancing RNA-targeting medicine requires comprehensive detection of variant-specific RNA-small molecule interactions and identification of genetic mutations that significantly affect their binding properties.

Technical limitations, however, have constrained previous studies to assessing only the binding against the reference genome while ignoring the effects of genetic variants. Conventional approaches rely

[1]Center for iPS Cell Research and Application, Kyoto University, Kyoto, Japan. [2]xFOREST Therapeutics Co., Ltd, Kyoto, Japan. [3]Institute of Multidisciplinary Research for Advanced Materials, Tohoku University, Miyagi, Japan. [4]Institute for Quantitative Bioscience, The University of Tokyo, Tokyo, Japan. ✉e-mail: onizuka@tohoku.ac.jp; saitou.hirohide.8a@kyoto-u.ac.jp; krk@xforestx.com

**Fig. 1 | Modification-induced RT deletions to detect RNA-small molecule interactions. a, b** Overview of BIVID-MaP. **a** Overview of binding-dependent modification by Rbs-VQ. Rbs (binding unit) binds to target RNA structures and then VQ (modification unit) covalently modifies the base near target RNA structures. **b** The binding site is detected by reverse transcription deletion (RT deletion) caused by binding-dependent modifications. **c** Chemical structures of alkylating modifiers used to probe RNA binding by berberine, CMA, and SMN-C2. **d** Gel shift assays showing the covalent modification by each modifier on the target RNA sequence. Berberine and CMA target G4 HIV-1 LTR and SMN-C2 targets Loop GAAGGAAGG. For visualization, SYBR Gold staining was used for the G4 HIV-1 LTR, and 5′-FAM labeling was used for the Loop GAAGGAAGG. From bottom to top, the gel shift bands show unmodified RNA, a 1:1 covalent complex (one modifier per RNA) and a 1:2 covalent complex (two modifiers per RNA) (See Supplementary Note). Representative data are shown from technical triplicate. **e** Modification-derived RT deletion frequency (RT-deletion), calculated as the deletion percentage with modifier minus the deletion percentage without modifier. Plots show RT-deletion at each nucleotide of the target binding site for each small molecule. The x-axis indicates nucleotide positions in G4 HIV-1 LTR for berberine and CMA, and in Loop GAAGGAAGG for SMN-C2. The target region is highlighted below each graph. **f** Model structures of the G-quadruplex in the HIV-1 LTR G4 and of the hairpin loop in Loop GAAGGAAGG, annotated with the RT-deletion (%) observed at each nucleotide. Target region of Loop GAAGGAAGG is highlighted with a dotted line. BIVID-MaP was performed in two independent experiments (n = 2). Source data are provided as a Source data file.

on reverse transcription termination (RT stop) near small-molecule binding sites[2,3]. Generated cDNA fragments lose sequence information beyond the RT stop, including upstream SNVs, thus making it difficult to assign sequencing reads to individual variants and distinguish variant-specific interactions. Mutational profiling approaches mitigate this problem by preserving full-length cDNA[7,8], but cannot clearly distinguish modification-induced base substitutions from target SNVs, thus limiting their scalability for variant detection. Indeed, no studies have approached variant-specific RNA-small molecule interactions to date.

Here, we introduce Binding- and Vinyl-Quinazolinone-Induced Deletion-Based Mutational Profiling (BIVID-MaP), a method capable of identifying variant-specific RNA-small molecule interactions. BIVID-MaP combines binding-dependent covalent modification with profiling of deletions upon reverse transcription (RT deletions) by deep sequencing to detect small-molecule binding sites (Fig. 1a, b). Furthermore, most deletions are single-nucleotide long, indicating that nearly all sequence information, including variant positions, is retained

despite these RT deletions. This feature distinguishes reads with different single-nucleotide variants, allowing the detection of deletions for each variant individually. Using this approach, BIVID-MaP discovered numerous cancer-associated somatic mutation-specific interactions between 5′ UTRs and a G-quadruplex (G4)-binding small molecule. Our results demonstrate that somatic mutations can alter target RNA structures in a position- and base-type-dependent manner and, consequently, influence the binding intensity of small molecules. Overall, BIVID-MaP provides a systematic approach to identify variant-specific RNA-small molecule interactions with single-nucleotide resolution.

## Results

### Detecting RNA-small molecule interaction by RT deletion profiling

To develop BIVID-MaP and detect variant-specific interactions, we examined combinations of covalent modifications and reverse transcriptases that could induce RT deletions (Fig. 1a, b). Previous research

on frameshift deletions induced by DNA polymerase indicates that planar modifications, such as 1,$N$2-etheno-dG, induce deletions by providing a larger stacking surface than canonical bases[9]. The π−π stacking interactions of the modified base with the terminal base of the elongating strand and the incoming dNTP are considered crucial for this process. Based on this mechanism, we hypothesized that a similar modification on RNA would induce RT deletions.

We previously reported a modifier comprising an RNA-binding small molecule (Rbs) conjugated to a dimethylamino-protected vinyl-quinazolinone, termed VQ (NMe$_2$)[10,11]. This Rbs-VQ (NMe$_2$) can be activated to the vinyl VQ form, enabling covalent bonding with bases, especially uridine bases (U), near a target RNA structure. The N3 position of U is the primary modification site. We investigated whether this VQ-modified U could induce deletions near the Rbs binding sites.

Our molecular modeling and MD simulation analyses of an RNA/cDNA complex containing a VQ-modified base suggest that the VQ moiety stacks with the terminal base of the growing cDNA strand (at the −1 position), thereby occupying the incoming nucleotide binding position (Supplementary Fig. 1a, b). This stacking persisted in the complex with Thermostable Group II Intron Reverse Transcriptase (TGIRT)[12], a commonly employed RT enzyme in conventional mutation profiling[8] (Supplementary Fig. 1c). We also observed a potential pocket for accommodating the Rbs and linker regions of the VQ modifier. To develop BIVID-MaP for detecting variant-specific interactions of small molecules, we first assessed how VQ-mediated covalent modification induces the three main types of RT mutations (i.e., substitution, deletion, and insertion).

As the first Rbs, we focused on berberine[13,14] because it selectively binds to G-quadruplex (G4) structures formed from guanine-rich sequences, which play an important role in gene expression regulation and are promising drug targets[15–18]. Berberine-VQ (NMe$_2$)[11] was synthesized and reacted with a known G4 structure from HIV-1 LTR[19] (Fig. 1c, d). Next, the frequencies of the three types of RT mutations were compared in the presence and absence of the modifier at each nucleotide. Interestingly, when we used TGIRT for cDNA synthesis, RT deletions at bases near the target G4 structure increased significantly (Fig. 1e, f). This increase was specific to deletions, as a similar increase was not observed for other mutation types (i.e., base substitutions or insertions) (Supplementary Fig. 3a). Moreover, SuperScript IV (SSIV) showed no significant change in any mutation types compared to TGIRT (Supplementary Fig. 3b). Thus, we discovered that the combination of VQ modification and TGIRT selectively promotes RT deletions. This discovery led us to develop a technique that detects RNA-small molecule interactions via binding-induced RT deletions.

To demonstrate the versatility of our method in using various small molecules, we tested other RNA-binding compounds in place of berberine. Replacing berberine with other RNA-binding small molecules still produced significant RT deletions near the target RNA structure. For example, VQ (NMe$_2$) compounds conjugated to another G4-binding small molecule, an acridine derivative (CMA)[10], generated a clear RT deletion signal at the target site (Fig. 1c–f). Furthermore, to demonstrate whether our method can detect binding to different types of RNA structures, we tested SMN-C2, which is an analog of an FDA-approved RNA-targeting drug for spinal muscular atrophy[20]. SMN-C2-VQ (NMe$_2$) treatment resulted in significant RT deletions in the target hairpin structure in Loop GAAGGAAGG[21] (Fig. 1c–f). Notably, the significant RT deletions were diminished when target purine-rich sequences were in the stem structure, meaning that our method recognizes the selective binding to the loop structure of SMN-C2 (Supplementary Fig. 4). These results demonstrate that the VQ-conjugated RNA-binding moiety can be replaced with other RNA-binding compounds.

We then verified that the observed RT deletions were due to covalent modification by VQ. As a control, the low-reactive Rbs-VQ (SMe)[10] showed no increase in RT deletions, in contrast to the highly reactive Rbs-VQ (NMe$_2$), thereby confirming that these deletions arise from covalent modification rather than from binding alone (Supplementary Fig. 5a, b). Additionally, RT deletions increased in a time-dependent manner, directly demonstrating progressive covalent modification by Rbs-VQ (Supplementary Fig. 5c). These findings establish that BIVID-MaP detects RNA-small molecule interactions through targeted covalent modification. Notably, regardless of which binding unit was employed, most deletions were one nucleotide in length (Supplementary Fig. 6). Thus, we confirmed that BIVID-MaP preserves the entire RNA sequence except for the deleted base, a critical and necessary feature for unambiguous assignment of sequencing reads to each variant[7,8].

## Reducing necessary read depth by adding a pull-down step

Moreover, to extend the analysis to larger scales and increase the sensitivity, we added a streptavidin-based pull-down step to enrich the modified RNAs and increase the RT deletion rate (Supplementary Note, Supplementary Fig. 8a). An azide-functionalized Berberine-VQ-N$_3$ (NMe$_2$) was synthesized for the click chemistry-based enrichment of modified RNAs (Supplementary Fig. 8b). Enrichment shows an 8.7-fold increase in RT-deletion for the G4 HIV-1 LTR and a significant increase in Σ RT-deletion for the target G4 sequence (Supplementary Fig. 8c, d). ROC curves, generated from random sampling of sequencing reads, revealed that pull-down samples require only 500 reads per gene for accurate binding detection (Supplementary Fig. 9a). This threshold is comparable to those employed by existing mutational profiling methods[7]. In contrast, samples without pull-down failed to reliably distinguish binding even with 5000 reads (Supplementary Fig. 9b). This finding indicates that including the pull-down step reduces the necessary sequencing read depth by at least 10-fold.

## Comparison between BIVID-MaP and other MaP methods

To evaluate the performance of BIVID-MaP relative to other mutational profiling methods, we conducted SHAPE-MaP[7,22,23] and DMS-MaPseq[8] in the presence and absence of RNA-binding ligands to compare mutation rates (Supplementary Fig. 10a). SHAPE-MaP failed to detect RNA-small molecule interactions identified by BIVID-MaP, as no significant differential RT-mutations were observed in positive controls compared to negative controls (Supplementary Fig. 10b, Supplementary Figs. 11–13). DMS-MaPseq missed some berberine-G4 structure interactions (G4 pre-mir-1229, G4 BCL2, G4 (UGGU)6, which are clearly detected by BIVID-MaP (Supplementary Fig. 10b, Supplementary Fig. 11). Furthermore, DMS-MaPseq could not detect SMN-C2 binding to GA-rich hairpin loops (Supplementary Fig. 13c, d). These method-dependent outcomes may reflect mechanistic differences between mutational profiling methods (Supplementary Note). Overall, these results demonstrate that BIVID-MaP provides superior sensitivity and specificity for detecting RNA-small molecule interactions in our validation system.

## BIVID-MaP detects the RNA-small molecule interaction modulated by an SNV

We then tested whether BIVID-MaP could detect changes in RNA-small molecule binding caused by a single-nucleotide mutation (Fig. 2a, b). We used an RNA from the *CD44* gene in which a G4 structure is disrupted by a single base change, specifically by replacing a guanine in the wild-type sequence (*CD44* I8 WT) with an adenine to create the mutant sequence (*CD44* I8 Mut)[24] (Fig. 2c). While circular dichroism (CD) analysis confirmed that *CD44* I8 WT forms a G4 structure with a typical CD pattern of RNA G4 that is dependent on the buffer conditions, the mutant *CD44* I8 Mut did not (Supplementary Fig. 14a). It was verified that only *CD44* I8 WT exhibited an increase in RT deletion events when each sequence was individually reacted with Berberine-VQ-N$_3$ (NMe$_2$) (Supplementary Fig. 14b).

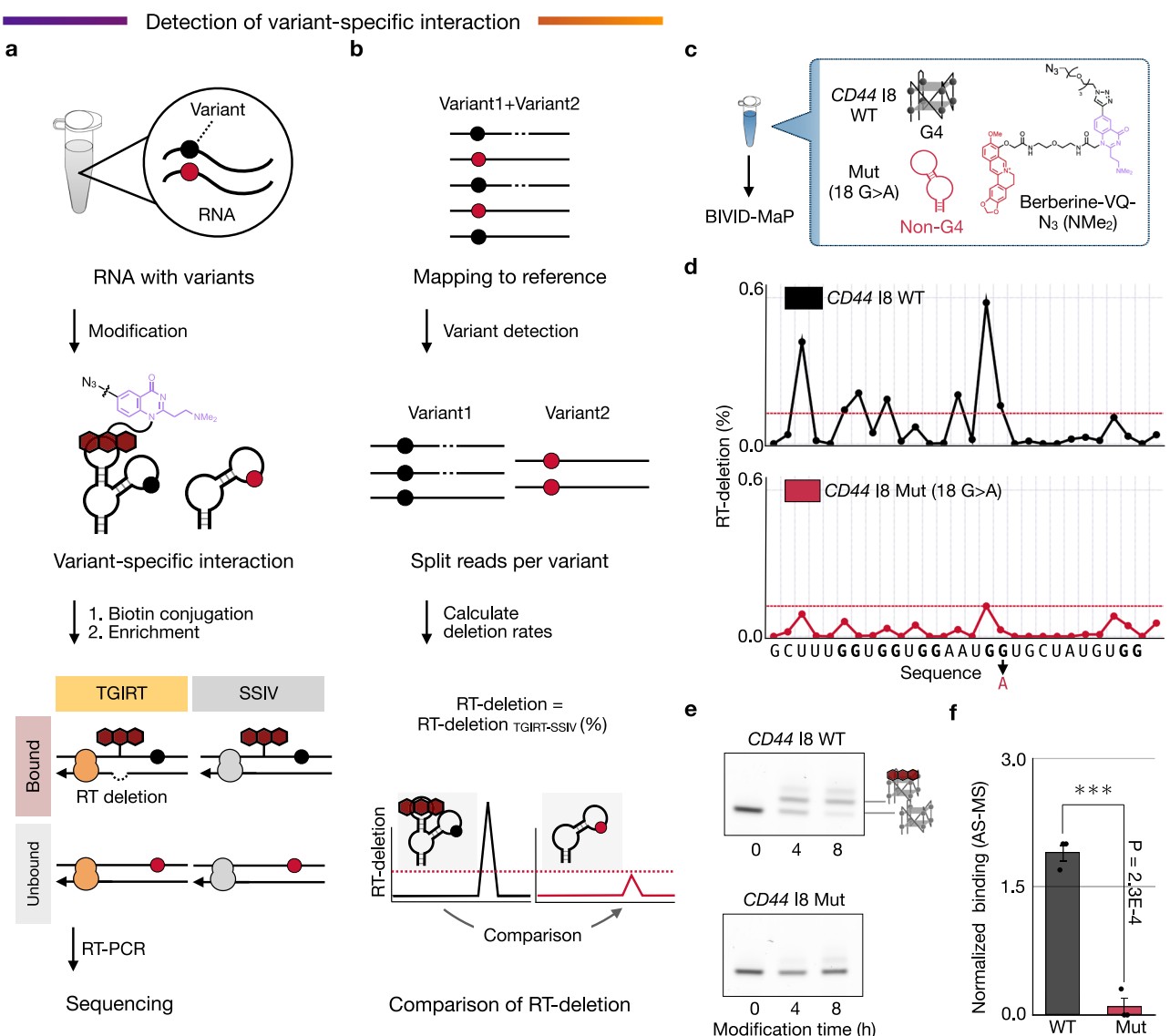

**Fig. 2 | Variant-specific interaction between berberine and the G4 structure disrupted by G to A mutation. a**, **b** Workflow for the detection of variant-specific interaction using BIVID-MaP. **a** A mixed RNA library containing variants is subjected to covalent modification by a small molecule. Variant-specific interactions lead to selective modification, which induces RT deletions near the modification site with TGIRT but not with SSIV. The cDNA is sequenced to identify deletion signatures corresponding to variant-specific RNA-small molecule interactions. **b** Sequence reads were mapped to the reference sequence, and each read was then classified by variant. The RT-deletion, representing the difference in RT deletion between TGIRT and SSIV conditions, is compared between variants to detect mutations affecting RNA-small molecule interactions. **c** Schematic of the model experimental condition to detect variant-specific interactions. BIVID-MaP was performed with WT and mutant RNAs mixed in one tube and reacted with Berberine-VQ (NMe$_2$) together. **d** RT-deletion profile for WT (black) and mutant (red). Red dashed line shows the highest RT-deletion in the mutant. The G-tract-forming guanines are highlighted in bold. BIVID-MaP was performed in two independent experiments (n = 2). **e** Gel shift analysis showing the covalent modification of Berberine-VQ (NMe$_2$) on each RNA variant (WT vs. mutant). From bottom to top, the gel shift bands show unmodified RNA, a 1:1 covalent complex (one modifier per RNA) and a 1:2 covalent complex (two modifiers per RNA) (See Supplementary Note). 5′-FAM labeled RNA was used for gel visualization. Representative data are shown from technical triplicate. **f** AS-MS analysis quantifies the amount of berberine binding to WT or mutant. MS signals are normalized to the mean intensity of each injection. Error bars indicate the means ± s.e.m. from three independent AS-MS measurements. Statistical significance was determined by two-sided Welch's $t$-test (***$p < 0.001$). Source data are provided as a Source data file.

Following this verification, BIVID-MaP with Berberine-VQ-N$_3$ (NMe$_2$) was applied to a mixture containing both *CD44* I8 WT and *CD44* I8 Mut RNAs, to mimic heterozygosity (Fig. 2c). Sequencing reads were separated, with RT deletions calculated for each sequence (Fig. 2b). RT deletion events were more frequent in *CD44* I8 WT RNA compared to *CD44* I8 Mut RNA (Fig. 2d). Additionally, gel-shift assays revealed that the G > A substitution markedly lowered the modification efficiency of Berberine-VQ-N$_3$ (NMe$_2$) (Fig. 2e). Consistent with this result, affinity-selection mass spectrometry (AS-MS)[25–27] showed a reduction in bound berberine (Fig. 2f). These results indicate that

BIVID-MaP can discriminate sequences differing by a single nucleotide and detect binding events specific to each variant. Indeed, BIVID-MaP was the only method among those evaluated that identified this SNV-specific binding, which was not replicated by either SHAPE-MaP or DMS-MaPseq (Supplementary Fig. 15).

## Large-scale analysis uncovers somatic mutations changing berberine binding to 5′ UTRs

RNA-binding small molecules that target G4 structures in oncogenic 5′ UTRs hold great promise for cancer therapy[28,29]. Over 30,000 cancer-

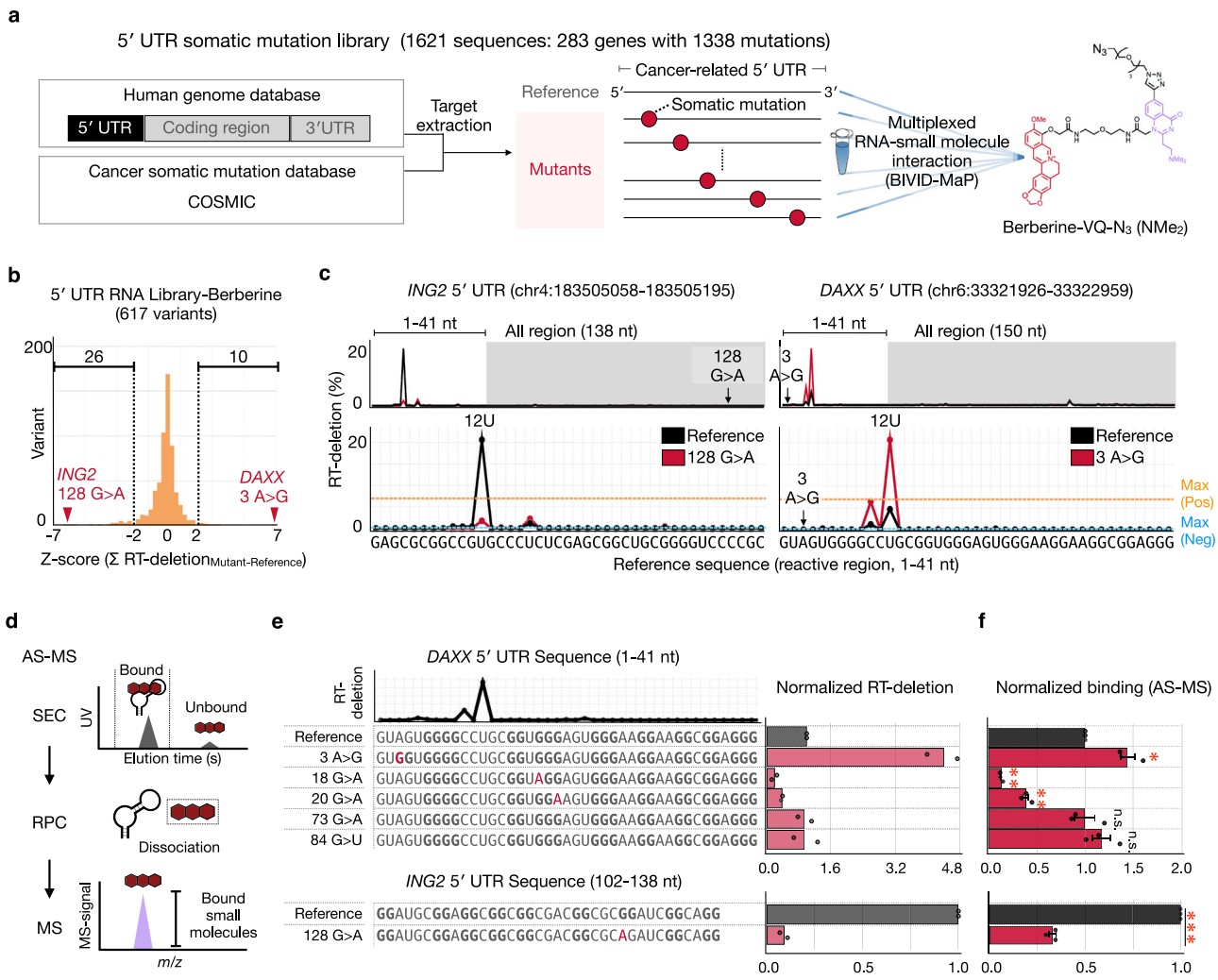

**Fig. 3 | High-throughput identification of variant-specific interactions between berberine and the 5′ UTR somatic mutation library. a** The 5′ UTR somatic mutation library contains each reference 5′ UTR and its mutants carrying the cancer-associated somatic mutation. The library was screened to detect variant-specific interactions with Berberine-VQ-N₃ (NMe₂). **b** Histogram of Z-score (Σ RT-deletion$_{Mutant-Reference}$) for each gene in the 5′ UTR RNA library screened with Berberine-VQ-N₃ (NMe₂). Σ RT-deletion is the sum of RT-deletion with *q*-value < 0.01 for each sequence. **c** RT-deletion profiles for the reference (black) and mutant (red) 5′ UTR. Overall reactivity is shown above, with 1–41 nt (containing the most reactive nucleotide) enlarged below. The highest peaks associated with the positive (G4 HIV-1 LTR) and negative control sequences (Non-G4 pre-mir-4520-1) are highlighted by orange or blue dashed lines, respectively. **d** The workflow of the affinity-selection mass spectrometry (AS-MS). After the target RNA is incubated with a VQ-free small molecule compound, the mixture is fractionated by size-exclusion chromatography

(SEC) into bound and unbound fractions. The bound fraction is then subjected to reverse-phase chromatography (RPC) to dissociate the complexes, and the mass spectrometer detects small-molecule signals from the RNA–bound fraction. **e** Normalized RT-deletion for each variant. The most reactive RT-deletion site in the reference was used for normalization. Bars represent values calculated from the mean read counts of two independent experiments (n = 2), with individual data points overlaid. The mutant nucleotide is highlighted in red, and G-tract-forming guanines are highlighted in bold. **f** Normalized binding of berberine to each variant detected by AS-MS. Error bars indicate the means ± s.e.m. of three independent experiments. Statistical significance was evaluated by two-sided Welch's *t*-test (***P = 2.5E-6 for *ING2* 128 G > A; **P = 4.7E-03 for *DAXX* 18 G > A, 8.5E-03 for *DAXX* 20 G > A; *P = 1.0E-2 for *DAXX* 3 A > G; n.s., P = 7.4E-1 for *DAXX* 73 G > A, 1.0E-1 for 84 G > U). BIVID-MaP was performed in two independent experiments (n = 2). Source data are provided as a Source data file.

associated somatic mutations have been identified in 5′ UTRs[30], with some mutations affecting RNA secondary structures[31]. To determine how such mutations influence small molecule recognition, we performed BIVID-MaP using Berberine-VQ-N₃ (NMe₂) and a large-scale RNA library containing 1621 5′ UTR sequences from 283 cancer-related genes. Each 5′ UTR includes a reference sequence from GENCODE[32] v44 and its mutant harboring a cancer-associated somatic mutation[33] (Fig. 3a, Supplementary Data 1). By comparing RT-deletion between each reference sequence and mutant sequences, we observed somatic mutations that significantly affect berberine binding.

We identified numerous somatic mutations that either increase or decrease berberine binding across multiple genes (Fig. 3b, Supplementary Fig. 16). The *DAXX* 3 A > G variant produced the greatest

increase in RT-deletion, and the *ING2* 128 G > A variant caused the largest decrease (Fig. 3b, c). Within the *DAXX* gene, 19 kinds of variants showed diverse effects on binding, depending on their position and type (Supplementary Fig. 17a, Supplementary Fig. 18a). These results demonstrate that even within a single 5′ UTR, somatic mutations can differentially influence small-molecule recognition.

To confirm whether the change in RT-deletion causes a corresponding change in berberine binding, we performed the AS-MS binding assay using VQ-free berberine (Fig. 3d). As expected, the 3 A > G variant, which increases RT-deletion in the *DAXX* gene, significantly increased berberine binding (Fig. 3e, f). By contrast, the 18 G > A and 20 G > A variants, which reduce the RT-deletion, significantly decreased binding. Similarly, in the *ING2* gene, the 128 G > A

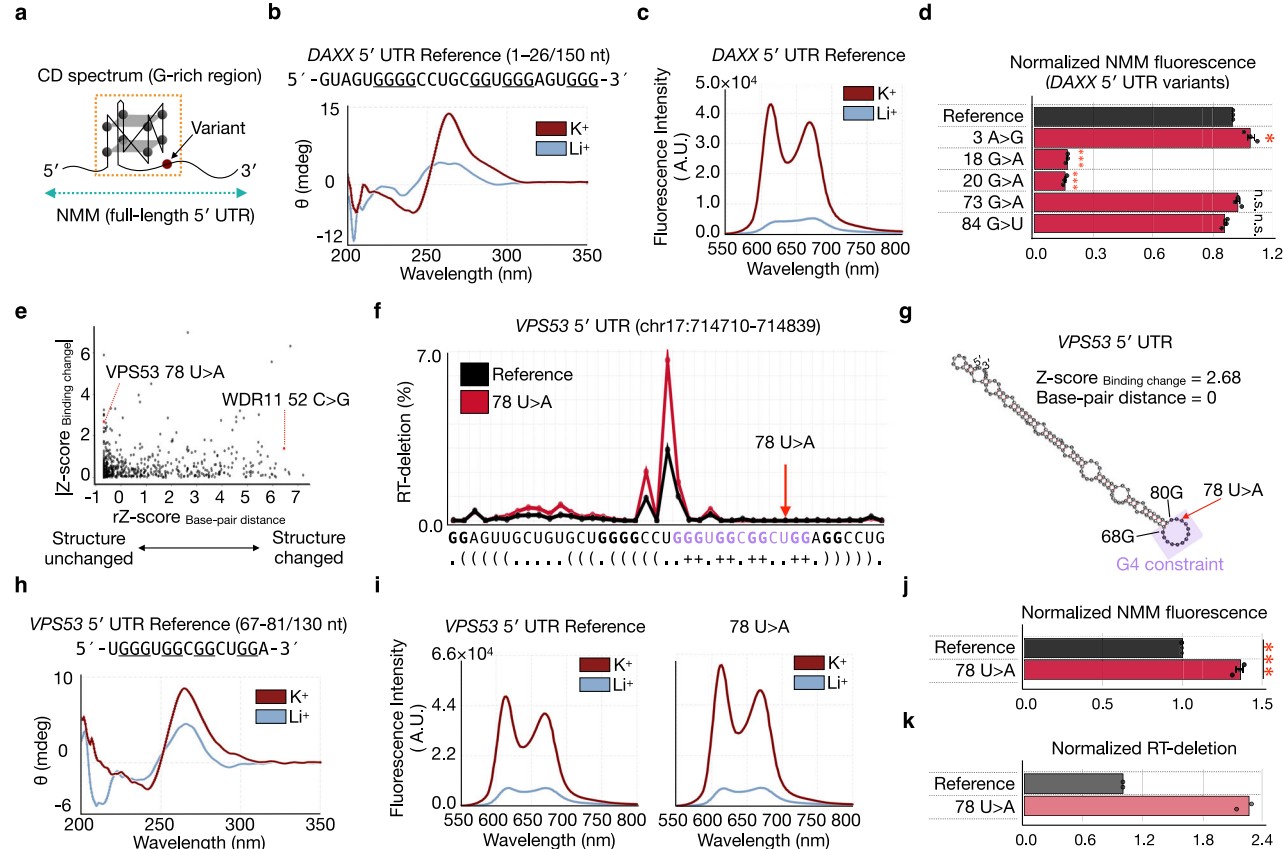

**Fig. 4 | Discovery of variant-specific G4 structure-dependent binding. a** To validate G4 structural formation, the CD spectra of the G-rich region of target 5′ UTRs were measured, and a G4 sensing assay with *N*-methylmesoporphyrin IX (NMM) was performed using full-length 5′ UTRs. **b** CD spectra of G-rich region (1–26 nt) of *DAXX* 5′ UTR reference in buffers containing either K+ or Li+. **c** NMM fluorescence spectra ($\lambda_{ex}$ = 399 and $\lambda_{em}$ = 550–800 nm) of full-length *DAXX* 5′ UTR Reference, showing weak signals under Li+ ions and strong signals under K+ ions, indicating G4 structure formation. **d** Normalized NMM fluorescence ($\lambda_{ex}$ = 399 and $\lambda_{em}$ = 610 nm) for each variant in *DAXX* 5′ UTR. Error bars represent means ± s.e.m. of three independent fluorescence measurements. Statistical significance was evaluated by two-sided Welch's *t*-test (\*\*\**P* = 1.1E-7 for 18 G > A, 7.5E-6 for 20 G > A; \**P* = 3.5E-2 for 3 A > G; n.s., *P* = 7.1E-2 for 73 G > A, 6.7E-2 for 84 G > U). **e** Correlation between SNV-induced RNA structural changes and changes in binding to small molecules. The x-axis represents the base-pair distance between the maximum expected accuracy (MEA) secondary structure of reference and mutant (normalized by 5′ UTR length), and the y-axis represents an indicator of variant-specific berberine binding, defined as Σ RT-deletion$_{Mutant-Reference}$. The larger the positive value on the x-axis, the greater the SNV-induced RNA structural changes. The larger

the positive value on the y-axis, the greater the change in binding. **f** RT-deletion profiles for the reference (black) and mutant (red) *VPS53* 5′ UTR. Sequences and secondary structure in dot-bracket are shown in the x-axis. G4 sequences are highlighted in purple. (+) indicates guanine residue forming a quadruple helix. RNA secondary structure was predicted using RNAfold with the option for G-quadruplex formation. **g** The predicted secondary structure of *VPS53* 5′ UTR. The G4 structure region is constrained as an unpaired loop and is highlighted in purple. **h** CD spectra of G-rich region (67–81 nt) of *VPS53* 5′ UTR reference in buffers containing either K+ or Li+. **i** NMM fluorescence spectra ($\lambda_{ex}$ = 399 and $\lambda_{em}$ = 550–800 nm) of full-length *VPS53* 5′ UTR Reference or 78 U > A variant. **j** Normalized NMM fluorescence ($\lambda_{ex}$ = 399 and $\lambda_{em}$ = 610 nm) for each variant in *VPS53* 5′ UTR. Error bars represent means ± s.e.m. of three independent fluorescence measurements. Statistical significance was evaluated by two-sided Welch's *t*-test (\*\*\**P* = 1.1E-5 for 78 U > A). **k** Normalized RT-deletion for each variant. The most reactive RT-deletion site in the reference was used for normalization. Bars represent values calculated from the mean read counts of two independent experiments (n = 2), with individual data points overlaid. BIVID-MaP was performed in two independent experiments (n = 2). Source data are provided as a Source data file.

variant, which resulted in the largest reduction in RT-deletion (Fig. 3c, Supplementary Fig. 17b), also decreased berberine binding significantly (Fig. 3e, f). These results showed that variant-induced changes in RT-deletion were consistent with changes in berberine binding. Additionally, VQ lacking the target small molecule (VQ-N₃) generated only low-level background signals and did not recapitulate the pronounced variant-specific peaks observed with Berberine-VQ-N₃ (NMe₂), supporting that the variant-specific RT-deletions primarily arise from target ligand-dependent binding (Supplementary Fig. 19b).

## Variants changing berberine binding affect G4 structural formation

We next examined how RNA structural changes caused by these somatic mutations relate to changes in binding. In *DAXX* 5′ UTR, the highest RT-deletion was observed at 12 U within a guanine-rich region

potentially forming a G4 structure (Fig. 3c, e, Supplementary Fig. 20). Binding-modulating variants cluster within the guanine-rich region, highlighting the importance of this region for berberine binding (Supplementary Fig. 17a).

To evaluate G4 formation and variant effects in the *DAXX* 5′ UTR, we performed CD spectroscopy and an *N*-methylmesoporphyrin IX (NMM) fluorescence assay, in which NMM shows selective fluorescence enhancement upon binding to G4 structures[34,35] (Fig. 4a). CD analysis of an extracted G-rich region (1–26 nt) confirmed K+-dependent G4 folding (Fig. 4b, Supplementary Fig. 21a), but did not reveal variant-dependent differences anticipated from berberine-binding, likely because the excised fragment does not fully preserve the original RNA structure.

We therefore extended the assay to the full-length 5′ UTR (150 nt). The *DAXX* 5′ UTR reference exhibited increased NMM fluorescence in

K[+] relative to Li[+], indicating the G4 formation in the full-length construct (Fig. 4c). Notably, NMM-enhanced fluorescence changed in a variant-specific manner concordant with berberine binding. The 3 A > G variant significantly increased fluorescence, whereas 18 G > A and 20 G > A variants significantly decreased it (Fig. 4d, Supplementary Fig. 22b). By contrast, variants outside the G-rich region (73 G > A, 84 G > U) had no effect. Together, these results indicate that a G4 within the DAXX 5′ UTR contributes to berberine interactions and that cancer-associated somatic mutations modulate the formation of the target G4 structure.

Additionally, to examine effects on other ligands that recognize G4 structures, we performed BIVID-MaP using CMA-VQ-N$_3$ (NMe$_2$). Variants in the DAXX 5′ UTR reproduced the same RT-deletion signature in 12U observed with Berberine-VQ (NMe$_2$) (Supplementary Fig. 19a). These results indicate that the cancer-associated somatic variants identified in DAXX can broadly modulate the binding of G4-recognizing ligands by affecting G4 structural formation.

Surprisingly, current in silico RNA secondary-structure prediction algorithms[36] cannot predict G4 structural changes induced by these somatic mutations. In the *DAXX* gene, the variants that significantly changed the berberine binding were located outside the predicted G4 structure-forming region (Supplementary Fig. 17a). Additionally, sequence-based G4 prediction software (cGcC[37], G4Hunter[38], G4NN[39]) showed only modest differences between variants and did not reproduce the pronounced variant-specific changes observed in berberine binding (Supplementary Fig. 20). These findings indicate that the BIVID-MaP method can elucidate non-canonical RNA structural changes and their effects on the small-molecule binding that are undetectable by RNA secondary structure predictions.

Similarly, the 128 G > A variant of the *ING2* 5′ UTR, which showed a reduction in RT-deletions, is predicted to contain G4-forming regions by multiple software tools (Supplementary Fig. 17b, Supplementary Fig. 20). The 12U is distant from the region containing the variant in sequence, but structurally close (Supplementary Fig. 23). Disruption of the G-tract by the 128 G > A variant markedly reduced RT deletions with both Berberine-VQ-N$_3$ (NMe$_2$) and CMA-VQ-N$_3$ (NMe$_2$), suggesting that this variant broadly affects ligands that recognize G4 structures (Fig. 3c, Supplementary Fig. 19a). However, neither the CD spectrum nor NMM fluorescence showed canonical G4 features (Supplementary Fig. 21b, Supplementary Fig. 22c). Thus, we could not confirm any clear G4 structure-dependence for the *ING2* 5′ UTR, raising the possibility that G4 formation is too weak to be detected by conventional G4 detection assays, or that these small molecules bind to non-G4 structures.

### SNV-induced RNA structural changes affect small-molecule binding

We further gained deeper insights into the relationship between SNV-induced RNA structural changes and their effects on small-molecule binding. We identified two contrasting variant patterns: mutations that affect berberine binding by changing the consensus RNA structure and those that do not (Fig. 4e).

The *WDR11* 52 C > G, a variant that modestly increased the RT-deletion for berberine, shows a broad reorganization of the predicted 5′ UTR secondary structure (Supplementary Fig. 24a, b). In the *WDR11* 5′ UTR, canonical G4 signatures were not observed by CD spectra nor NMM fluorescence (Supplementary Fig. 21d, Supplementary Fig. 22d), or sequence-based G4 predictors (Supplementary Fig. 20). Additionally, CMA-VQ (NMe$_2$) also showed an elevated RT-deletion at 52 C > G (Supplementary Fig. 19a). One possible explanation for the absence of a clear G4 signature is intercalation of the small molecule into the RNA, as planar cationic aromatics are known to partially intercalate into non-G4 RNAs[40,41]. Under this hypothesis, the increased signal is more likely to reflect non-G4 structural reorganization rather than G4 formation.

In the case of the *VPS3* 5′ UTR, although the 78 U > A variant is predicted to maintain the consensus RNA structure of the reference sequence, it resulted in a significant increase in berberine binding (Fig. 4f, g). Since a G4 structure-forming region is present at 68–80 nt (Fig. 4f, Supplementary Fig. 20), we focused on the Watson-Crick base-pair probability that competes with Hoogsteen base pairing required for G4 folding[42–44]. The Watson-Crick base-pair probability in the target G4 structure region is lower in the 78 U > A variant (Supplementary Fig. 24c), possibly weakening canonical Watson-Crick pairing within the G4 structure, thereby affecting G4 structure formation. This observation indicates that, even when variants do not alter the predicted consensus structure, they can reshape the ensemble of RNA substructures by changing the base-pair probabilities within the target motif, thereby modulating small-molecule binding.

Consistent with this interpretation, the guanine-rich regions of both the reference and the 78 U > A variant displayed G4 structure-specific CD spectra (Fig. 4h, Supplementary Fig. 21c). Importantly, 78 U > A significantly enhanced NMM fluorescence, confirming increased stabilization of G4 structures by this mutant (Fig. 4i, j). This change is consistent with the observed increase of RT-deletion with Berberine-VQ-N$_3$ (NMe$_2$), showing that strengthened G4 formation underlies the higher berberine binding (Fig. 4k). Notably, binding was affected in a G4 structure-dependent manner, even though the 78 U > A substitution does not alter the G-tracts that directly influence quadruplex formation[17], and sequence-based G4 prediction remained unchanged (Supplementary Fig. 20). This result highlights the utility of BIVID-MaP in discovering variant-specific RNA structure-small molecule interactions that are difficult to predict solely from sequence information. Moreover, the absence of a comparable variant-specific increase in the RT-deletion with CMA-VQ-N$_3$ (NMe$_2$), relative to Berberine-VQ-N$_3$ (NMe$_2$), suggests that the effect of 78 U > A may be dependent on the target G4-binding small molecule (Supplementary Fig. 19a).

In summary, BIVID-MaP revealed that cancer-associated somatic mutations modulate the 5′ UTR-small molecule interactions by affecting target RNA structure formation in multiple genes.

## Discussion

Although SNVs are known to alter RNA structures dramatically, the effects on RNA-small molecule interactions have remained unexplored[4,5]. In this study, we introduce BIVID-MaP, a high-throughput sequence-based method using binding-dependent covalent modification and RT deletion profiling for systematic detection of variant-specific RNA-small molecule interactions. By capturing RT deletion signatures at single-nucleotide resolution, BIVID-MaP reveals how distinct genetic variants modulate ligand binding, providing insights for evaluating RNA-binding small molecules.

BIVID-MaP fills a crucial gap in our understanding of RNA-ligand recognition. While many small molecules have been developed to target RNA structures[1], most studies have focused on interactions with reference genomes, overlooking the potential influence of genetic variation. Yet, recent studies have found that branaplam, discontinued from Huntington's disease trials, promotes unique splicing modulation patterns in patient-derived cells carrying SNVs[6], emphasizing the need to assess both on-target and off-target effects from genetic variations in the development of RNA-targeting compounds. Our approach directly addresses this issue by enabling the detection of SNV-specific binding events. BIVID-MaP paves the way for designing RNA-targeting compounds that account for the effects of individual genetic variation.

Here, we discovered that the combination of VQ and TGIRT leads to modification-derived RT deletions. Notably, these deletions are mostly one nucleotide in length, ensuring that nearly the entire sequence, including variant positions, is retained. This high resolution enables unambiguous discrimination of reads with SNVs and variant-specific deletion profiling to quantify binding changes. Compared to

other mutational profiling methods[7,8], BIVID-MaP is unique as it specifically detects RT deletions. For two key reasons, BIVID-MaP employs RT deletions instead of RT substitutions. First, RT deletions are more sensitive and specific than RT substitutions for detecting binding sites. In our berberine-G4 interaction detection, RT substitutions frequently produced false-positive signals in repetitive nucleotide regions (Supplementary Fig. 25), whereas such non-specific signals were not observed in deletion-based analysis. Second, detecting nucleotide substitutions carries a risk of misinterpreting the variant of interest as mutations induced by chemical modifications. RT deletions, however, clearly distinguish single-nucleotide substitution variants. Therefore, BIVID-MaP could provide a more accurate means to detect binding specific to single-nucleotide substitution variants.

The utility of BIVID-MaP was further demonstrated through the large-scale screening of a 5′ UTR library containing patient-derived cancer-associated somatic mutations. By comparing RT deletions between reference and mutant sequences, we identified several somatic mutations that significantly modulate small-molecule binding. For example, in the *DAXX* gene, the 3 A > G variant increased binding, whereas the *ING2* 128 G > A variant decreased it. These findings highlight how genetic diversity can lead to previously unrecognized interactions that dynamically change the binding of small molecules. Additionally, BIVID-MaP separately detected small-molecule binding to SNVs at different positions within the same gene. The changes in RT deletion were consistent with the changes in binding from AS-MS measurement with VQ-free berberine. Rigorous evaluation of RNA-small molecule binding by BIVID-MaP may help reassess existing compounds and optimize drug discovery for individuals with specific genetic variants.

A limitation of BIVID-MaP is that it cannot directly distinguish between variants when the RT deletion and variant sites overlap. In such cases, a secondary assay is required to examine each variant independently to assess its deletion profile separately. This limitation is mitigated when analyzing multiple SNVs in the same sequence, for example, when SNPs cluster due to linkage disequilibrium[45,46], meaning that even if one SNV is lost due to a deletion event during RT, adjacent linked SNVs are likely to remain detectable (Supplementary Fig. 26).

Regarding RNA-modifying moieties, there remains scope for optimization. As previously reported, VQ exhibits high selectivity towards U bases. Consequently, this method missed the berberine-G4 interactions when sequences completely lack U (G4 pre-mir-6850, Supplementary Figs. 7, 11). Conversely, even when U is absent within the target RNA structure itself, structurally proximate U residues outside the motif may still serve as reaction sites, which could partially mitigate this bias. From this perspective, the Rbs-VQ linker requires appropriate length and flexibility to enable modification of structurally adjacent U residues. Furthermore, prior studies on frameshift deletions[9] and our MD simulations support VQ as one of the suitable moieties that provides a stacking surface and can efficiently induce RT deletion (Supplementary Fig. 1). Future work introducing modifiers with sufficient stacking ability while targeting other bases could extend this approach into a less biased platform.

Our research demonstrated that BIVID-MaP can detect RNA-small molecule interactions with $K_d$ values ranging from sub-micromolar to low micromolar ($K_d = 0.47\,\mu M$ for the berberine-G4 TERRA interaction[11] and $K_d = 4\,\mu M$ for the SMN-C2-Loop GAAGGAAGG interaction[21]). The range of detectable binding affinities depends on the compound concentration used in the experiment and the affinity of each RNA-small molecule pair. Increasing the compound concentration improves modification efficiency, thereby extending the range of affinities that can be detected by BIVID-MaP towards weaker interactions.

Considering the broader application of BIVID-MaP, fast-reacting modification chemistry is well-suited to the dynamic and unstable RNA

structural environment in cells[47–49], where G4 structures, in particular, are often unfolded under complex cellular contexts[50]. Importantly, we demonstrated that pre-activation of the VQ-precursor by a 4-h incubation with methyl vinyl ketone (MVK) enables rapid modification, even when the reaction time is shortened to 10 min (Supplementary Fig. 27a, b). During this pre-incubation, MVK facilitates VQ generation through the irreversible capture of the eliminated secondary amine. This process shifts the chemical equilibrium toward the formation of the active VQ species and prevents the reverse reaction, thereby bypassing what was previously the rate-limiting elimination step. This pre-activation allowed us to detect berberine-G4 interactions in HeLa cell lysates with sensitivity similar to that observed in the buffer, confirming that a 10-min incubation is sufficient for detecting berberine binding (Supplementary Fig. 27c−e). A possible concern is that glutathione in the cellular environment might react with VQ, which could suppress its modification efficiency. In our reaction system, MVK not only accelerates the reaction but also minimizes undesired reactions with intracellular thiols by acting as a trapping reagent. This enables efficient modification even under conditions that mimic the complex cellular environment, thus suggesting the feasibility of extending BIVID-MaP to intracellular applications, although further optimization of the VQ-conjugated compounds, including improvements in membrane permeability, is still needed.

BIVID-MaP can be applied to a broad range of genetic variations, including SNPs and rare SNVs, which are known to affect RNA structure and function[4,5]. Due to the high scalability by adding the enrichment step of modified RNA using an azide-conjugated modifier, BIVID-MaP is an ideal tool for a comprehensive evaluation of how different mutations affect RNA-small molecule interactions through large-scale screening. Moreover, BIVID-MaP shows great promise in discovering RNA structures, particularly those altered by single-nucleotide mutations, known as riboSNitches[4,5]. By using berberine, which recognizes G4 structures, we were able to identify potential G4 formations and somatic mutations that can affect their stability. Importantly, the SNV-dependent binding differences detected by BIVID-MaP may result from both intrinsic structural variations and differential structure induction by ligand binding. Small molecules like berberine can induce and stabilize specific RNA conformations[11,51,52] and single nucleotide changes may alter the propensity for such compound-induced structural transitions. Therefore, the observed changes in binding patterns reflect not only pre-existing structural differences but also conformational changes caused by small-molecule recognition.

Overall, applying BIVID-MaP to any RNA-binding small molecule that recognizes specific RNA structures could enable the discovery of functional RNA structures and the assessment of mutation-induced structural changes. With insights gained into individual genetic variation-specific RNA-small molecule interactions, BIVID-MaP represents a powerful platform for advancing RNA structural biology and RNA-targeting drug discovery.

## Methods
### Linker design and probe synthesis
VQ-conjugated probes were synthesized as described previously[10,11] and in the Supplementary Methods. For linker selection, we primarily adopted PEG spacers as we expect PEG linkers to (i) provide sufficient reach and conformational flexibility for the VQ electrophile to access nearby uracils, (ii) maintain aqueous solubility, and (iii) reduce aggregation compared with purely alkyl linkers. In the design of SMN-C2, a longer linker was incorporated to enhance the interaction of SMN-C2-VQ with the hairpin loop structure and facilitate access to the distant U base. Specifically, an alkyl linker was introduced near the binding site to promote hydrophobic interactions, followed by a PEG linker connecting it to the VQ. The optimal linker length is crucial for efficient and unbiased detection, as linkers that are too short may result in insufficient modification yield.

### Design of target RNAs (excluding the 5′ UTR somatic mutation library)

Each single-stranded DNA sequence used as RNA template was designed with the following segments from the 3′ end: a reverse primer sequence for PCR amplification, a spacer (5′-TTT-3′), a universal stem sequence A (5′-TTCAGC-3′), the antisense DNA sequence corresponding to the target RNA structure, a universal stem sequence B (5′-GCTGAA-3′), another spacer (5′-TTT-3′), and a forward primer sequence for PCR amplification. Details of DNA and RNA sequences are available in Supplementary Data 2, 3.

### Design of the 5′ UTR somatic mutation library

Target genes were selected from ONGene[53] (a literature-based database of human oncogenes) and TSGene 2.0[54] (a database of tumor suppressor genes). Corresponding transcripts were chosen from the MANE (Matched Annotation from NCBI and EMBL-EBI) database[55]. 5′ UTR sequences were downloaded from the UCSC Table Browser[56] using GENCODE v44[32], including only genes with 5′ UTR length ≤182 nt. Somatic single-nucleotide mutations in these regions were retrieved from the COSMIC database[33]. For each gene, we prepared the reference 5′ UTR sequence and mutant 5′ UTR sequences with COSMIC mutations. Each DNA template (used as the RNA template) was designed (from the 5′ end) with a forward primer sequence for PCR, the target 5′ UTR sequence, and a reverse primer sequence for PCR. Primer sequences were chosen to form a stem-loop structure for preserving the target secondary structure. DNA templates were purchased from Twist Bioscience. Details of DNA and RNA sequences are available in Supplementary Data 1.

### DNA template amplification

Before in vitro transcription, a double-stranded DNA template was prepared via PCR amplification of the DNA template. PCR was performed in a 25 μL reaction containing 0.5 μL of Platinum SuperFi II DNA Polymerase (Thermo Fisher Scientific), 1× SuperFi II Buffer (Thermo Fisher Scientific), 500 nM forward primer, 500 nM reverse primer, and the single-stranded DNA template. For each primer set, both the annealing temperature and the number of cycles were optimized (Supplementary Data 2, 3, 6). The reaction mixture was initially heated to 98 °C for 30 s, followed by an optimized number of cycles at 98 °C for 10 s, the optimized annealing temperature for 10 s, and 72 °C for 15 s. After cycling, the reaction was held at 72 °C for 5 min and then cooled to 4 °C. After amplification, the double-stranded DNA product was purified using a Monarch PCR & DNA Cleanup Kit (New England Biolabs).

### In vitro transcription

For RNA probe preparation or library generation, in vitro transcription was performed using the MEGAshortscript T7 Transcription Kit (Thermo Fisher Scientific). A 20 μL reaction containing the double-stranded DNA template was incubated at 37 °C for 18 h. After transcription, 2 μL TURBO DNase (Thermo Fisher Scientific) was added, mixed by pipetting, and incubated at 37 °C for 15 min to digest the DNA template. The RNA product was then purified using an RNA Clean & Concentrator kit (Zymo Research).

### Target RNA modification

The target RNA was first folded by adding 1× PKN buffer (20 mM phosphate, 80 mM KCl, 20 mM NaCl, pH 7.0), heating at 95 °C for 5 min, and then cooling on ice. Next, 1 μM folded RNA was incubated in a 20 μL reaction mixture containing 1× PKN buffer and 1 mM EDTA with an Rbs-VQ precursor (NMe₂ or SMe). The final DMSO concentration was 10% (v/v). The VQ precursor concentration of 4 μM was used for berberine or CMA, whereas 40 μM was employed for SMN-C2. The standard reaction time was 8 h for berberine or CMA and 28 h for SMN-C2 at 37 °C. For negative controls with non-azide VQ compounds, DMSO was added instead of the Rbs-VQ precursor. After incubation, the RNA was purified using an RNA Clean & Concentrator kit (Zymo Research).

### Click reaction

For each 300–500 ng aliquot of modified RNA, 2 μL of 2 mM Click-iT Biotin sDIBO Alkyne (Thermo Fisher Scientific) and 1 μL RiboLock RNase Inhibitor (Thermo Fisher Scientific) were added, and nuclease-free water was added to bring the volume to 30 μL. The mixture was incubated at 37 °C for 2 h with shaking (1000 rpm) in an Eppendorf Thermomixer. Afterward, the RNA was purified using an RNA Clean & Concentrator-5 kit (Zymo Research).

### Pull-down after click reaction

20 μL of SpeedBeads Magnetic Neutravidin Coated Particles (Cytiva) were placed in a 1.5 mL tube on a magnetic rack, and the supernatant was removed. Next, 1000 μL Binding and Wash buffer (5 mM Tris-HCl pH 7.5, 0.5 mM EDTA, 1 M NaCl, 1% Tween 20) was added and the tube was gently inverted to mix. The beads were washed three times by magnetically pelleting and removing the supernatant. The beads were then washed twice with 1000 μL Solution A (0.1 M NaOH, 0.05 M NaCl) and once with 1000 μL Solution B (0.1 M NaCl) in the same manner[57]. After washing, the click-treated RNA sample was added to the bead pellet, and additional Solution B was added to bring the volume to 200 μL. The mixture was incubated at 4 °C for 2 h with shaking (1200 rpm). Beads were then magnetically collected, with the supernatant removed. Beads were washed three times with 1000 μL Binding and Wash buffer. Finally, 50 μL Elution Buffer (95% formamide, 10 mM EDTA, pH 8.2) was added to the beads and heated at 80 °C for 5 min. The sample was then cooled to room temperature for 5 min, placed on the magnet, and the supernatant containing the eluted RNA was transferred to a new tube. The RNA was finally purified using an RNA Clean & Concentrator-5 kit (Zymo Research).

### Reverse transcription

Reverse transcription was performed using a primer complementary to the RNA's 3′-terminal sequence. First, a 10 μL primer annealing mixture (7 μL modified RNA, 1 μL 2 μM reverse primer, 2 μL 10 mM dNTPs) was prepared and incubated to anneal the primer. A 2.22× RT buffer was freshly prepared, consisting of 2.22× MaP pre-buffer (5× MaP pre-buffer: 250 mM Tris-HCl pH 8.0, 375 mM KCl, 50 mM DTT), 2.22 M betaine, and 11.1 mM MgCl₂. Reverse transcription was then performed using the following program: 25 °C for 10 min, 60 °C for 90 min, 90 °C for 10 min, then held at 4 °C. For TGIRT-III (Ingex), the 20 μL reaction contained 0.5 μL of TGIRT-III enzyme and 0.5 μL of TGIRT dilution buffer (20 mM Tris-HCl pH 7.5, 400 mM KCl, 50% glycerol), 9 μL 2.22× RT buffer, and 10 μL annealed RNA mixture. For SuperScript IV (SSIV), the 20 μL reaction contained 4 μL 5× SSIV buffer (Thermo Fisher Scientific), 4 μL 100 mM DTT, 1 μL RNase OUT inhibitor (Thermo Fisher Scientific), 1 μL SSIV reverse transcriptase, and 10 μL annealed RNA. After cDNA synthesis, 1 μL RNase H was added and the mixture was incubated at 37 °C for 20 min to degrade the RNA. The cDNA was then purified using an Oligo Clean & Concentrator-5 kit (Zymo Research).

### Preparation of Illumina sequencing library

To prepare libraries for Illumina sequencing, a two-step PCR (amplicon PCR followed by index PCR) was performed. For the amplicon PCR, a 25 μL reaction containing cDNA, 0.5 μL of Platinum SuperFi II DNA Polymerase, 1× SuperFi II Buffer, and 500 nM forward and reverse primers was first heated to 98 °C for 30 s, followed by cycles of 98 °C for 10 s, 64 °C for 10 s, and 72 °C for 20 s. After cycling, the reaction was held at 72 °C for 5 min and then cooled to 4 °C. Next, 2.5 μL Exonuclease I (New England Biolabs) was added and the mixture was incubated at 37 °C for 15 min to digest residual primers. The product was

purified with a Monarch PCR & DNA Cleanup kit (New England Biolabs) and eluted in 8 µL of nuclease-free water. For the index PCR, a 25 µL reaction containing amplicon PCR product, 0.5 µL of Platinum SuperFi II DNA Polymerase, 1× SuperFi II Buffer (Thermo Fisher Scientific), and 0.5 µM each of Nextera XT Index Kit v2 index primers (Illumina) was prepared. The mixture was heated to 98 °C for 30 s, followed by 7 cycles of 98 °C for 10 s, 55 °C for 10 s, and 72 °C for 20 s. After cycling, the reaction was held at 72 °C for 5 min and then cooled to 4 °C. The indexed library was purified using Sera-Mag Select beads (Cytiva). For elution, 14 µL nuclease-free water was added to the dried beads, mixed thoroughly, incubated at room temperature for 10 min before collecting the supernatant. Finally, samples with distinct index sequences were pooled for sequencing.

### Sequencing
Sequencing was performed on an Illumina platform using either an iSeq 100 i1 Reagent v2 kit (300-cycle) or a NovaSeq 6000 SP Reagent Kit v1.5 (300-cycle), with paired-end reads and standard Illumina primers.

### Alignment and data analysis
Adapter sequences were trimmed from the raw reads (FASTQ files), with reads aligned to reference sequences using BWA[58]. The deletion frequency at each nucleotide position was calculated as the number of reads with a deletion at that position divided by the total read count. To reduce noise from intrinsic sequence biases, the deletion rate in an untreated (DMSO) sample was subtracted from that in the Rbs-VQ treated sample to obtain RT-deletion (%), defined by Equation (1).

$$RT - deletion = RT - deletion_{Rbs-VQ} - RT - deletion_{DMSO} \quad (1)$$

Additionally, to avoid biases introduced by the enrichment step (which can alter the sequence distribution in the library), a control sample reverse-transcribed with SSIV (which does not induce deletions) was used for background correction when enrichment was performed. In this case, deletion frequency was defined by Equation (2). After Rbs-VQ modification, the deletion rates in TGIRT and SSIV samples were denoted as RT-deletion$_{Rbs-VQ, TGIRT}$ and RT-deletion$_{Rbs-VQ, SSIV}$, respectively.

$$RT - deletion = RT - deletion_{Rbs-VQ, TGIRT} - RT - deletion_{Rbs-VQ, SSIV} \quad (2)$$

The binding change associated with SNV is quantified by Equation (3), where n is the length of each sequence.

$$\sum RT - deletion_{Mutant-Reference} = \sum_{k=1}^{n} RT - deletion_{Mutant} - \sum_{k=1}^{n} RT - deletion_{Reference} \quad (3)$$

The statistical significance (p-value) of RT-deletion at each base was assessed by the right-tailed Fisher's exact test. The p-values for each RNA were adjusted by the Benjamini-Hochberg procedure to obtain q-values. RT-deletion values for bases with $q < 0.01$ were then summed to calculate Σ RT-deletion, representing the binding for each sequence. A minimum read depth of 1000 was required for each sequence in the analysis. The binding change associated with an SNV was normalized as a Z-score using Equation (4). $\mu$ is the mean value of the library population, $\sigma$ is the standard deviation, and $x$ is each binding change quantified by Equation 3.

$$Zscore_x = \frac{x - \mu}{\sigma} \quad (4)$$

Because the base-pair distance data were non-normally distributed, we applied a robust Z-score using Equation (5), in which represents the set of base pair distance data, median(X) as the median, and is the median absolute deviation.

$$rZscore_x = \frac{x_i - median(X)}{MAD(X) \times 1.4826} \quad (5)$$

For the analysis of variant-specific interactions, sequencing reads are first mapped to the reference sequence for each RNA, generating a SAM file that contains all variants. Variants are then identified individually, and a separate SAM file is produced for each variant. Each variant-specific SAM file is subsequently used to calculate the RT-deletion.

In SHAPE-MaP and DMS-MaPseq, RT mutations incorporating all three types of variation (substitution, deletion, and insertion) were employed. In this case, the mutation frequency is defined by the following Equation (6). The mutation rate under conditions with ligand and modifier is denoted as RT-mutation$_{Ligand+, Modifier+}$, and the mutation rate under conditions with no ligand but with modifier is denoted as RT-mutation$_{Ligand-, Modifier+}$.

$$RT - mutation = RT - mutation_{Ligand+, Modifier+} - RT - mutation_{Ligand-, Modifier+} \quad (6)$$

### Confirmation of modification by gel shift assay (Except for G4 HIV-1 LTR)
Target RNAs were purchased from JBioS (Japan), GeneDesign. An RNA construct was prepared with a common stem sequence at both ends (5′ end: 5′-AGC-3′; 3′ end: 5′-GCU-3′) flanking the target RNA structure. This common stem sequence was not added to the Loop GAAGGAAGG construct, as its target structure is inherently stabilized by the stem. For gel visualization, RNAs were labeled with 5′-FAM (fluorescein) at the 5′ terminus. This construct was used to assess the modification rate of Rbs-VQ via a gel shift assay. First, the RNA was heated at 95 °C for 5 min in 1× PKN buffer and then cooled to 4 °C to allow folding. Next, a 20 µL reaction mixture containing 1 µM RNA, 1× PKN buffer, 4 µM of each Rbs-VQ precursor, and 1 mM EDTA was prepared. The final DMSO concentration was 10% (v/v). Reactions with the berberine or CMA precursors were incubated for 4 or 8 h, whereas those with the SMN-C2 precursor were incubated for 8 or 28 h. For berberine and CMA, azide-labeled modifiers were employed in all sequences to facilitate visualization of gel shifts. Samples were mixed with loading buffer prior to electrophoresis and loaded onto either (i) a 16% denaturing polyacrylamide gel (prepared in TBE buffer with 20% formamide), where an aliquot of the reaction mixture was mixed with an equal volume of loading buffer and electrophoresed at 500 V for 90–100 min at room temperature, or (ii) a 15% Novex TBE Urea Gel (Thermo Fisher Scientific), where 1 µL of the reaction mixture was mixed with 9 µL of loading buffer and electrophoresed at 200 V for 80–90 min. Gels were imaged for FAM fluorescence using a ChemiDoc Touch MP system (Bio-Rad) or Typhoon FLA 7000 (GE Healthcare), with band intensities quantified using Image Lab software (Bio-Rad). Unmodified RNA incubated for 0 h served as a mobility reference for the gel shift. The detailed RNA sequences, along with the type of gel used for each sequence and the fluorescence detection system, are listed in Supplementary Data 4.

### Confirmation of modification by gel shift assay for G4 HIV-1 LTR
For the G4 HIV-1 LTR RNA (64 nt), the longest sequence in which we performed gel shift assays, the adduct size was increased post-reaction to facilitate detection of gel mobility shifts. Non-FAM-labeled RNA was purchased from Agilent. RNA design, folding, and primary modification were performed as described above. For modification, an

azide-attached Rbs-VQ precursor was used. The resulting azide-modified RNA (10 μL) was subjected to a click reaction by adding 2 μL of 50 μM DBCO-PEG4-amine and incubating at 37 °C for 3 h. Next, a 12 μL aliquot of the solution was mixed with 12 μL of loading buffer. A 6 μL sample of this mixture was then loaded onto a 16% denaturing polyacrylamide gel (containing 20% formamide in TBE buffer) and subjected to electrophoresis at 500 V for 90–100 min at room temperature. After electrophoresis, RNA was stained with SYBR Gold and imaged using a ChemiDoc Touch MP system (Bio-Rad). Band intensities were quantified using Image Lab software (Bio-Rad). Unmodified RNA incubated for 0 h served as a mobility reference for the gel shift. Detailed RNA sequences are provided in Supplementary Data 4.

## CD-spectrum analysis
All RNAs were synthesized and purchased from IDT. RNA samples (2.0 μM) were prepared in 1× K⁺ folding buffer (10 mM Tris-HCl pH 7.5, 100 mM KCl) or 1× Li⁺ folding buffer (10 mM Tris-HCl pH 7.5, 100 mM LiCl). All RNA samples (150 μL) were incubated at 95 °C for 5 min and then cooled to 4 °C for at least 5 min. Before the analysis, the samples were incubated at room temperature for at least 5 min. CD spectra were acquired on a J-720WI (JASCO) with a Peltier temperature controller set to 25 °C. All samples were analyzed in a micro quartz cell with a 1 cm path length. The data were averaged over five scans with subtraction of the buffer baseline. Details of the RNA sequences are available in Supplementary Data 5.

## Affinity selection mass spectrometry (AS-MS)
The target RNA (4 μM) was first heated at 95 °C for 5 min in a 1× PKN buffer, then cooled to 4 °C to allow folding. Next, a 20 μL reaction containing 2 μM folded RNA, 1× PKN buffer, and berberine chloride (Funakoshi) was incubated at 4 °C for 1 h. The final DMSO concentration was 10% (v/v). The concentration of berberine chloride was 8.8 μM for the *DAXX* 5′ UTR and 200 μM for the *ING2* 5′ UTR and CD44 I8. Affinity selection mass spectrometry (AS-MS) was performed using a system configured according to the automated ligand identification system (ALIS) approach[26]. The system consisted of an Agilent 1260 Infinity II quaternary pump (G7111B), multisampler (G7167A), two variable wavelength detectors (G7114A), and valve drive (G1170A), coupled to an Agilent InfinityLab LC/MSD single quadrupole mass spectrometer (G6125BA). Size-exclusion chromatography (SEC) was performed using a PolyHYDROXYETHYL A column (50 × 2.1 mm, 3 μm, 60 Å; PolyLC). Reversed-phase chromatography (RPC) was conducted using an Agilent InfinityLab Poroshell 120 EC-C18 column (3.0 × 50 mm, 1.9 μm). For the SEC step, 15 μL of the sample was injected onto a SEC column using 1× PKN buffer as the mobile phase at a flow rate of 400 μL/min at 4 °C. Then, the UV detector monitors the eluent at 260 nm absorbance to extract the SEC fraction containing both unbound RNA and RNA-ligand complexes. The extracted RNA-ligand complexes was then directed into the RPC by the valving system. In the RPC at 60 °C, the ligand dissociated from the RNA-ligand complex with a linear gradient from 2 to 98 % of acetonitrile in 0.1% formic acid was eluted into a mass spectrometer at 500 μL/min. Mass spectrometric detection was performed in selected ion monitoring (SIM) mode, targeting $m/z$ 336.12 corresponding to berberine. The system was controlled, and data were analyzed using OpenLab ChemStation software (Agilent). Details of the RNA sequences are available in Supplementary Data 6.

## RNA structure prediction and calculation of Watson-Crick base-pair probabilities
RNA secondary structures were predicted by the RNAfold software using the ViennaRNA v2.5.0 package with the option for G-quadruplex formation (command: RNAfold -g --MEA --noPS). Watson-Crick base pairing probabilities for each nucleotide were obtained by computing the partition function with ViennaRNA default parameters. For each

nucleotide i, the probabilities p(i,j) (j ≠ i) were summed across the corresponding row of the resulting base-pair probability matrix using a custom Python 3.8.12 script. To draw RNA structures containing sequences predicted to form G4 structures, the ViennaRNA "constraint" option was employed, forcing the putative G4 tract to remain unpaired during secondary-structure prediction[59].

## RNA-small molecule interaction detection by SHAPE-MaP
The modification and reverse transcription protocols were based on a previous publication[60]. First, the target RNA was folded by adding 1× PKN buffer (20 mM phosphate, 80 mM KCl, and 20 mM NaCl at pH 7.0), followed by heating at 95 °C for 5 min and cooling on ice. Next, 1 μM of the folded RNA was incubated in a 20 μL reaction mixture containing 1.1× PKN buffer, 1 mM EDTA, and either an Rbs-VQ precursor or a VQ-free compound. In addition to the VQ-free compound, we also performed SHAPE-MaP using Rbs-VQ (SMe), which has low modification reactivity, to account for potential effects of the VQ moiety itself. Non-G4 structures were used as less reactive controls for berberine. The VQ-free compounds Berberine-chloride (TCI), 9′-Amino-6-chloro-2-methoxyacridine (Funakoshi, referred to as CMA), and SMN-C2 (TargetMol) were purchased from their respective vendors. A concentration of 4 μM was used for berberine or CMA, whereas 40 μM was used for SMN-C2. As a control, DMSO solutions were added instead of the compound. The RNA-small molecule mixture was incubated at 37 °C for 60 min. Then, 9 μL of the RNA-small molecule mixture was added to 1 μL of a 500 mM 2A3 solution in DMSO to give a final reaction volume of 10 μL and a final 2A3 concentration of 50 mM. This was then incubated at 37 °C for 10 min. After incubation, the reaction was quenched by adding 10 μL of 500 mM DTT. The modified RNA was then purified using 40 μL of RNAclean XP beads (Beckman Coulter).

For reverse transcription, first, a 10 μL primer annealing mixture (7 μL of modified RNA, 1 μL of a 2 μM reverse primer, and 2 μL of 10 mM dNTPs) was prepared and incubated to anneal the primer. Freshly prepared 2.22× SHAPE RT buffer consisted of 2.22× MaP pre-buffer (5× MaP pre-buffer: 250 mM Tris-HCl, pH 8.0; 375 mM KCl; 50 mM DTT), 2.22 M betaine, and 13.3 mM MnCl₂. Then, the annealed RNA (10 μL) was incubated with 1 μL SSII and 9 μL of the 2.22× SHAPE RT buffer. Reverse transcription was then performed using the following program: 25 °C for 10 min, 42 °C for 90 min, [50 °C for 2 min → 42 °C for 2 min] × 15, 72 °C for 10 min, then 4 °C. Then, 5 μL of a 1 M NaOH and 0.4 M EDTA solution was added, and the mixture was incubated at 95 °C for 3 min to hydrolyze the RNA. The mixture was neutralized by the addition of 1 μL of 5 M HCl solution and then purified using 52 μL of RNAcleanXP. The sequence library was subsequently prepared using the same protocol as that described for BIVID-MaP in the preceding sections.

## RNA-small molecule interaction detection by DMS-MaPseq
The target RNA was folded by adding 1× PKN buffer (20 mM phosphate, 80 mM KCl, and 20 mM NaCl at pH 7.0), followed by heating at 95 °C for 5 min and cooling on ice. Next, 1 μM of the folded RNA was incubated in a 40 μL reaction mixture containing 1.03× PKN buffer, 1 mM EDTA, and either an Rbs-VQ precursor or a VQ-free compound. Ligands were purchased from the same vendors and used at the same concentrations as described for the SHAPE-MaP experiments. In addition to VQ-free compound, we also performed DMS-MaPseq using Rbs-VQ (SMe), which has low modification reactivity, to account for potential effects of the VQ moiety itself. As a control, DMSO was added instead of the compound. The RNA-small molecule mixture was incubated at 37 °C for 60 min. Then, 39 μL of the RNA-small molecule mixture was added to 1 μL of DMS to give a final reaction volume of 40 μL and a final DMS percentage of 2.5 %. This was then incubated at 37 °C for 2 min. After incubation, the reaction was quenched by adding 20 μL of 2-mercaptoethanol. The RNA was finally purified using an RNA

Clean & Concentrator-5 kit (Zymo Research). Subsequent library preparation, including reverse transcription, was performed using the same method as described for BIVID-MaP in the preceding section.

### ROC curve analysis

Random subsets of 100, 500, 1000, or 5000 paired reads were extracted three times from every SAM file with a custom Python script. Each downsampled SAM was then processed to calculate the RT-deletion for each sequence, yielding $\Sigma$ RT-deletion. Replicate $\Sigma$ RT-deletion profiles were aggregated, designating six G4 targets as true positives and five non-G4 targets as negatives (listed in Source Data), and these scores were used to generate ROC curves and AUCs under pull-down+ and pull-down- conditions to compare the sequencing depth required for berberine-G4 binding detection.

### Pre-activation of Berberine-VQ-precursor

For MVK pre-activation, reactions were prepared in 0.5 mL DNA LoBind tubes (Eppendorf). A mixture containing 14.25 µL 2× PKN buffer, 8.55 µL nuclease-free water, 0.15 µL 200 mM MVK in DMSO, and 1.5 µL Berberine-VQ-$N_3$ ($NMe_2$) was assembled to obtain a final concentration of 1× PKN buffer. The reaction was incubated at 37 °C for 4 h to generate MVK-pre-activated VQ probes. The pre-activated probe was used directly in subsequent alkylation reactions.

### Modification reaction in cell lysate

Modification in cell lysate was performed in 0.5 mL DNA LoBind tubes in a total volume of 30 µL. First, 1.5 µL of HeLa whole cell lysate (2.5 µg/µL; Santa Cruz Biotechnology, Inc.) was mixed with 0.75 µL SUPERase·In™ RNase Inhibitor (Thermo Fisher Scientific) and 1.5 µL RNase inhibitor (NIPPON GENE), followed by incubation at 37 °C for 4 h. In the buffer condition, an equal volume of nuclease-free water was added instead of cell lysate. Then, 0.3 µL of 200 mM MVK was then added, and the mixture was incubated at 37 °C for 10 min. Subsequently, 1.5 µL of 50 µM RNA in 1× PKN buffer and 24.45 µL of MVK-pre-activated VQ probe were added. The reaction was incubated at 37 °C for 10, 30, or 60 min to allow covalent alkylation. After incubation, samples were directly analyzed by denaturing PAGE or proceeded to sequencing library preparation using the same method as described in the preceding sections.

### Molecular dynamics simulations

All-atom molecular dynamics (MD) simulations were performed using OpenMM[61], with systems prepared via AmberTools[62]. The initial structure of TGIRT-RNA/cDNA ternary complex was modeled by MacroModel based on the reported crystal structure [6AR1][63]. Irrelevant chains were removed from the input structure to improve visual clarity. Ligand parameters for the small molecule VQ were generated with Antechamber (AmberTools) under the GAFF2 force field. Proteins, RNA, and DNA were assigned the ff19SB, OL3, and bsc1 force fields, respectively. Missing force field parameters were supplemented with parmchk2. System assembly was carried out in tleap (AmberTools). Default mbondi3 radii was used, consistent with implicit solvent models. The RNA fragment and TGIRT protein were loaded alongside the ligand, and a manual covalent bond was introduced between the N3 atom of RNA uracil and the C1 atom of VQ ligand. The resulting RNA-VQ-TGIRT complex was parameterized and output as Amber prmtop and inpcrd files, as well as a PDB reference file, for downstream MD simulations. Simulations were carried out in implicit solvent (GBn2) with a 0.10 M salt concentration. Nonbonded interactions were computed using a 1.0 nm cutoff under non-periodic boundary conditions. Bonds to hydrogen atoms were constrained using the SHAKE algorithm, permitting a 2 fs timestep following initial equilibration. The system was minimized and equilibrated under the NVT ensemble using a Langevin thermostat set to 300 K. The simulation protocol included an initial 10 ps equilibration at 300 K, followed by cooling to 277 K for ~100 ps, and subsequent heating back to 300 K for 100 ps. Energies were reported every 1000 steps, with trajectory coordinates saved every 1000 steps in both DCD and PDB formats. Analyses were conducted on the trajectory for a total of 210 ps.

### N-Methylmesoporphyrin IX (NMM) fluorescence assay

N-Methylmesoporphyrin IX (NMM, Frontier Scientific) was purchased and resuspended in a 1× K$^+$ folding buffer or 1× Li$^+$ folding buffer. All RNAs were generated by in vitro transcription as stated in the previous paragraph or purchased from IDT. RNA samples (4.0 µM) were prepared in 1× K$^+$ folding buffer or 1× Li$^+$ folding buffer. For folding, RNA was incubated at 95 °C and cooled to 4 °C at −6 °C/s on a ProFlex Thermal Cycler. Refolded RNA was then folded at 4 °C for at least 30 min. Next, 18.7 µL of refolded RNA (4.0 µM) and 11.3 µL of NMM (26.5 µM) were mixed, resulting in a final concentration ratio of RNA:NMM = 1:4. Fluorescence scanning was carried out using Infinite (TECAN) with an excitation wavelength of 399 nm and an emission wavelength range of 550–800 nm. The target fluorescence intensity was subsequently measured on the same Infinite (TECAN) instrument, with an excitation wavelength of 399 nm and an emission wavelength of 610 nm. The fluorescence intensity was corrected by subtracting the intensity of the buffer mixture as a background. Details of the RNA probes are available in Supplementary Data 7.

### Reporting summary

Further information on research design is available in the Nature Portfolio Reporting Summary linked to this article.

## Data availability

The data supporting the findings of this study are available from the corresponding authors upon request. The BIVID-MaP, SHAPE-MaP, and DMS-MaPseq sequencing data generated in this study have been deposited in the NCBI Sequence Read Archive (SRA) database under accession code PRJNA1392914. The oncogene and tumor suppressor gene annotation data used in this study are available in the ONGene[53] and TSGene 2.0[54] databases. The somatic mutation data used in this study are available in the COSMIC database (v98)[33]. The transcript data used in this study are available in the MANE database (v1.3)[55]. The 5′ UTR sequence data used in this study are available in the GENCODE database[32] (v44) accessible via the UCSC Table Browser[56]. The source data generated in this study are provided in the Source Data file. Source data are provided with this paper.

## Code availability

Custom scripts for classifying sequencing reads and calculating RT deletions per variant are available on GitHub with instructions for their use (https://github.com/BIVID-MaP/BIVID_MaP2025) and are archived on Zenodo (https://doi.org/10.5281/zenodo.18374906)[64].

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

## Acknowledgements

This work was supported by the Japan Society for the Promotion of Science (JSPS) KAKENHI (Grant Numbers JP20H05626 (H.S.), JP25H00970 (H.S.), JP23H02076 (F.N.), JP24K01641 (K.O.), JP23KJ1331 (E.M.)), the Japan Science and Technology Agency (JST) CREST (Grant Number JPMJCR23B3 (H.S.)), JST FOREST program (Grant Number JPMJFR2002 (K.O.)), iPS Cell Research Fund from Centre for iPS Cell Research and Application (Kyoto University).

## Author contributions

E.M. and K.R.K. conceived and designed the study. E.M. developed the BIVID-MaP analytical pipeline. K.O., Y.C., H.Y., S.I., and T.H. designed the modification reagents. K.O., Y.C., and H.H. developed the MVK-based modification reaction. K.O. and P.Y. performed molecular simulations. E.M., Y.C., H.Y., H.H., S.I., T.H., M.O., and K.M. performed experiments and analyzed the data. K.O., F.N., H.S., and K.R.K. supervised the study. E.M., K.O., H.Y., P.Y., H.S., and K.R.K. wrote the paper with advice from all authors.

## Competing interests
