## [Transparent Peer Review file · Nature Communications]

Systematic identification of variant-specific RNA structure-small molecule interactions exemplified by RNA G-quadruplexes

Corresponding Author: Dr Kaoru Komatsu

Version 0:

Reviewer comments:

Reviewer #1

(Remarks to the Author)

In this manuscript, the authors propose a new method, Binding- and Vinyl-Quinazolinone-Induced Deletion-Based Mutational Profiling (BIVID-MaP), to identify RNA–small molecule interactions. BIVID-MaP leverages the covalent RNA labeling ability of Vinyl-Quinazolinone (VQ) and utilizes small molecules conjugated with VQ to systematically map small-molecule binding sites on RNA.

The authors focus on the property that Thermostable Group II Intron Reverse Transcriptase (TGIRT) tends to induce single-nucleotide deletions at VQ-modified sites. They claim that BIVID-MaP can detect small-molecule binding sites that change in response to single-nucleotide mutation in RNAs

Using berberine, a small molecule known to bind RNA G-quadruplex structures, as a model compound, the authors demonstrate that berberine binding to a somatic mutation RNA library in the 5'-UTR region is affected by specific single-nucleotide variants.

Overall, the presented method is highly promising and creative, integrating covalent RNA labeling with sequencing-based profiling to explore RNA–small molecule interactions. To fully realize its potential, particularly in drug discovery contexts, some clarifications and further validation would help strengthen the manuscript.

As a reviewer, I believe that this work could merit publication in Nature Communications if the following concerns are adequately addressed. Otherwise, the work may still make a valuable contribution in a more specialized journal focused on chemical biology or RNA technologies.

[Major Comments]

1. Clarifying the Significance Relative to Existing Methods

As the authors themselves mention, several alternative techniques exist for mapping RNA–small molecule interactions, such as Chem-CLIP [J. Am. Chem. Soc. 144, 22 11620–11625 (2022)] and mutational profiling methods like SHAPE-MaP [Methods 167, 105–116 (2019)] and DMS-MaPseq [Nat. Methods 14, 75–82 (2017)].

While the authors briefly mention alternative approaches, a more detailed discussion of how BIVID-MaP complements or outperforms existing methods (e.g., Chem-CLIP, SHAPE-MaP) would help readers appreciate the unique value of this method. A direct comparison using the same RNA–ligand system could provide especially compelling support. For example, can BIVID-MaP identify RNA interaction sites with berberine that are not detectable by SHAPE-MaP or DMS-MaPseq? Or does it offer improved binding site resolution?

2. Generality Across RNA Structures

In this study, the authors synthesized and evaluated VQ-conjugated berberine, CMA, and SMN-C2. However, Figures 2 and subsequent figures primarily focus on berberine binding to RNA G-quadruplex structures.

Although Figures 2–4 show binding site identification for berberine–G-quadruplex interactions, the broader applicability of this method to other RNA structural motifs remains unclear.

VQ's covalent reactivity is known to exhibit nucleobase (e.g., toward thymine-N3, NAR 2019, 47, 6578) selectivity.

Furthermore, it is possible that the reactivity varies depending on the angle of the π^* orbital. Therefore, its labeling efficiency may not be uniform across diverse RNA structural contexts. In contrast, photoaffinity labeling with diazirines seems to offer less biased reactivity, which may lead to more comprehensive mapping.

Demonstrating that VQ can form covalent bonds at multiple binding sites, regardless of nearby base composition and RNA structures (at least some G-quadruplex structures or sequences), would support its broader utility. Does the flexibility of the

linker contribute to reducing selectivity?

3. Generality Across Small Molecules

This study largely centers on berberine-G4 interactions. As mentioned in Comment 2, the reactivity of VQ seems to be more selective than that of photoaffinity labels.

To support generalizability, the authors should investigate interactions between RNA and additional small molecules (e.g., CMA or SMN-C2) and compare the results with those obtained from other profiling methods, such as SHAPE-MaP or DMS-MaPseq.

If further validation with other small molecules is not currently feasible, the abstract, introduction, discussion, and conclusion should be narrowed to focus specifically on the berberine-G-quadruplex or G-quadruplex. Alternatively, using therapeutically more relevant RNA-binding lead compounds could clarify the significance of compound-specific profiling, even if generalizability is lower.

4. [Supplementary] Potential for Cellular Application

This is a supplementary comment and does not require additional experiments.

SHAPE-MaP and DMS-MaPseq can be performed in living cells. If BIVID-MaP could also be adapted for in vivo use, it would significantly enhance its relevance to drug discovery.

However, the membrane permeability of VQ-conjugated compounds may be a concern. A discussion of this point in the manuscript would help readers better understand the potential and limitations of this method.

[Minor Comments]

1. P.8, Lines 18–20 et al.:

As discussed, this method may enable the identification of structural changes in RNA resulting from single-base mutations at a level that structural prediction tools cannot predict. However, compounds like berberine often induce and stabilize specific RNA conformations upon binding.

Therefore, any observed changes in binding patterns should not be attributed solely to structural alterations of RNA in the absence of the compound. The authors should clearly state that such changes may result from structure induction upon small-molecule binding.

2. Extended Data Figure 1 (Berberine/TGIRT):

The positions of observed deletions and substitutions differ, with substitutions appearing more widespread.

The manuscript should clarify the interpretation of these patterns.

If deletions are prioritized as binding indicators while substitutions are ignored, the authors should provide a rationale to justify this analytical focus.

3. Extended Data Figure 1 (CMA and SMN-C2/TGIRT):

Substitutions are also observed with CMA and SMN-C2. However, their impact on analysis is not discussed.

The authors should clarify whether these substitutions could lead to false positives or otherwise affect interpretation.

Reviewer #2

(Remarks to the Author)

In this study, Miyashita et al. developed BIVID-MaP, a high-throughput method for detecting RNA-small molecule interactions that combines binding-dependent covalent modification with profiling of deletions upon reverse transcription via deep sequencing. From the data, they revealed numerous variant-specific interactions between an RNA-binding small molecule and RNA harboring single-nucleotide variants. In addition, several cancer-associated somatic mutations significantly influence the binding intensity of a small molecule by affecting target RNA structures. The overall data may contribute to the development of RNA-targeting drugs in the future.

Overall, this work potentially fits the aim and scope of Nature Communications. The authors have tried to address a research gap that is often gone unnoticed and have come up with a novel and practical solution to understand/report it. There are also individual examples and significance shown in this work, as they have demonstrated that it is linked to cancer-associated somatic mutations. However, there are some additional experiments and controls that are needed to support the claims of this manuscript, as well as to enhance the quality of data and the impact/implication of the study. The author also needs to perform some of these experiments in cellular context/lysate to showcase the usefulness of the method, as RNA structure in vitro and in cells can be very different, and the presence of other components may affect the chemical binding and crosslinking efficiency. I hope the authors will make an effort to revise the manuscript substantially and accordingly, which is required to meet/exceed the high standard of this prestigious journal. Once the authors can fully address the major and minor comments, the scientific quality of the manuscript will be substantially improved and I will be happy to re-review it.

Major comments :

1. Figure 1. It seems that the small molecule will mostly interact with U nearby the target binding region. This will lower the chance of crosslinking. Did they try other crosslinking groups or consider other strategies?

2. Figure 1. In addition, the crosslinking time is very slow, taking 8 or more hours. As RNA is quite unstable in vitro and in cells, and these chemicals may also be somewhat toxic. Has the authors demonstrated the dosage and time-dependent treatment on RNA degradation and cell viability, as well as other off-target binding and crosslinking in a cellular context/lysate?

3. Figure 1. It seems the linker is different for each case; however, it is not described in the maintext at all. It is suggested to show the data in results, or at least discuss the design of linker somewhere, as I assume it is also important for the binding and crosslinking efficiency, as it should be flexible enough to find the Us nearby. It is unclear at the moment on the design and whether others have been tested. The linker may also play an important role in cellular application, for solubility and cell permeability later.

4. Figure 1. The % detection seems very low, even though the gel showed ~50% crosslinking. Does it mean the TGIRT is

still not doing a very good job in deletion during RT? This also implies that relatively deep sequencing is needed to have a good signal-to-noise level. Can the author also comment on the sequencing requirement for the non-pull down and pull down case? This may also be a limitation of the method.

5. Figure 2. The authors should show that the G->A mutation can disrupt the G4 formation by some biophysical/biochemical assays, such as CD, UV melting, NMM staining gel, etc. A comparison to WT is needed. Please also do so for other individual candidates presented in other figures.

6. Figure 3. For the 2 examples shown, it is hard to understand how the 128 G>A affects the very upstream U. It is better to show the secondary structure or where the G4 structure is.

7. Figure 3. Did they verify the data with the other G4 ligand or no G4 ligand control (just linker+crosslinking part)? It is important to have these controls.

8. Figure 4. The authors may use RNAfold, which has an option to allow prediction of G4. Also, it may be good to put the sequence into some of those G4 prediction software, such as G4NN, G4 hunter. In addition, it seems hard to know where the G4 formation is and where the G4 ligand binding sites are. Again, the data should be verified with other G4 ligand and no G4 ligand control (just linker +crosslinking part)

Minor comments :

1. Figure 2. The G4-region should be bolded or underlined for easier identification. The same applies to other figures.

2. Why do the authors not directly incorporate biotin into the chemical, but use azide? Is it better for synthesis or later cell treatment/application?

3. There are multiple crosslinking bands seen on the gel, and the authors have not explained their origin and whether they will affect the library preparation and analysis.

4. It is not clear to me sometimes the author uses % deletion on the y-axis, and sometimes BIVID signal on the y-axis.

Reviewer #3

(Remarks to the Author)

The manuscript by Miyashita et al. describes the application of Binding- and Vinyl-Quinazolinone (BVQ)-based chemical probes previously reported for investigating G-quadruplexes induced or lost in 5' untranslated regions (UTRs) in mRNAs through single-nucleotide variants (SNVs). BVQ probes contain a latent electrophile, that upon binding to an RNA motif-of-interest, are activated for covalent labeling. In this work, BVQ probes are used in concert with a deletion-based mutational profiling method to identify labeling sites in a library of in vitro transcribed 5' UTR sequences from cancer-related genes that contain SNVs. Using a berberine-based probe and the developed "Binding- and Vinyl-Quinazolinone-Induced Deletion-Based Mutational Profiling" (BIVID-MaP) strategy, the team uncovered several SNVs that affect how berberine binds to the RNA (i.e., if a G-quadruplex structures was induced or disrupted due to the mutation). This later aspect is the most exciting part of the work with the potential of uncovering the presence of potentially druggable G-quadruplexes in cancer-relevant RNAs. While the overall concept is exciting for the field, there are several aspects of the manuscript that need improvement before publication.

1. The rate of reactivity of the BVQ probes is really slow (up to 28 hours!). Would this approach ever be applicability in a cellular context? While investigating in vitro transcribed RNAs is useful, many complexities of the cell may preclude formation of a structure, especially RNA G-quadruplexes which have been demonstrated to be fleeting, especially in 5' UTRs that are unwound by the ribosome. Discussion of this aspect should be added, as well as the potential biological relevance of the discovered motifs. There is major concern that many of these findings will not translate to cells, as has been observed with other RNA G-quadruplexes (e.g., in the NRAS transcript).

2. Using the TGIRT for deletions rather than mutations is a very interesting finding, yet biochemical details regarding why this enzyme can do this are missing. Additional details should be provided regarding this unique reactivity, particularly since the method relies heavily on this previously unknown reaction.

3. Discussion of how affinity affects labeling should be provided. Based on the slow kinetics of the reaction, is there an affinity threshold that must be met?

4. In several of the RNA gels, it appears that the RNA is being degraded over time (e.g., SMN-C2 in Figure 1C and CD44 in Figure 2E). Are the RNAs being degraded or can the RNA no longer be detected? Details regarding visualization of the RNA gels is missing in the methods. Is a dye being used that can no longer bind following probe labeling?

5. Again, regarding the gels, it is surprising that a detectable gel shift is observed considering the size of the RNAs. The authors should provide evidence that labeling stoichiometry is 1:1 (some seem like additional bands are formed over time). Along similar lines, probe to RNA equivalents should be added to the figure legend.

6. The title and abstract imply much greater impact with respect to SNVs than is described in the text. From my reading, the manuscript is focused almost solely on G-quadruplexes, and as such, the title and abstract should be rewritten to be more accurate. The manuscript largely describes a method (BIVID-MaP) that can be used to probe for G-quadruplex-containing RNAs; however, questions remain regarding the method due to the weak characterization of the TGIRT reactivity as described above in point 2.

Version 1:

Reviewer comments:

Reviewer #1

(Remarks to the Author)

The authors have provided appropriate and satisfactory responses to the reviewer's comments, including a comparison with competing technologies. This reviewer considers the manuscript suitable for publication in Nature Communications.

Reviewer #2

(Remarks to the Author)

The authors have done an excellent job in addressing all my comments. I therefore recommend the publication of the manuscript in Nature Communications.

Major comments :

1. OK.
2. OK.
3. OK.
4. OK.
5. OK.
6. OK.
7. OK.
8. OK.

Minor comments :

- 1-4. All OK.

Reviewer #3

(Remarks to the Author)

In this revised manuscript, the authors have largely addressed the comments from the previous critiques. In doing so, however, additional points to be addressed have been identified that will be outlined below:

1. Like the abstract, the title should also be updated to include G-quadruplex specificity.
2. How does MVK accelerate the reaction? Additional details should be provided or this should be removed and the subject of a future manuscript.
3. I find it concerning that SHAPE-MaP and DMS-MaP were not able to detect any RNA-small molecule interactions identified using BIVID-MaP. Have the authors identified other RNA-ligand systems that worked for all 3 methods. More comments regarding this lack of overlapping detection should be included in the discussion.

Open Access This Peer Review File is licensed under a Creative Commons Attribution 4.0 International License, which permits use, sharing, adaptation, distribution and reproduction in any medium or format, as long as you give appropriate credit to the original author(s) and the source, provide a link to the Creative Commons license, and indicate if changes were

made.

Reviewer 1

In this manuscript, the authors propose a new method, Binding- and Vinyl-Quinazolinone-Induced Deletion-Based Mutational Profiling (BIVID-MaP), to identify RNA-small molecule interactions. BIVID-MaP leverages the covalent RNA labeling ability of Vinyl-Quinazolinone (VQ) and utilizes small molecules conjugated with VQ to systematically map small-molecule binding sites on RNA.

The authors focus on the property that Thermostable Group II Intron Reverse Transcriptase (TGIRT) tends to induce single-nucleotide deletions at VQ-modified sites. They claim that BIVID-MaP can detect small-molecule binding sites that change in response to single-nucleotide mutation in RNAs. Using berberine, a small molecule known to bind RNA G-quadruplex structures, as a model compound, the authors demonstrate that berberine binding to a somatic mutation RNA library in the 5'-UTR region is affected by specific single-nucleotide variants.

Overall, the presented method is highly promising and creative, integrating covalent RNA labeling with sequencing-based profiling to explore RNA-small molecule interactions. To fully realize its potential, particularly in drug discovery contexts, some clarifications and further validation would help strengthen the manuscript.

As a reviewer, I believe that this work could merit publication in Nature Communications if the following concerns are adequately addressed. Otherwise, the work may still make a valuable contribution in a more specialized journal focused on chemical biology or RNA technologies.

Response:

We appreciate the reviewer's positive feedback and insightful comments that greatly helped enhance our manuscript. As recommended, we conducted additional experiments to substantiate the BIVID-MaP methodology. Furthermore, we incorporated new text regarding comparisons with other established methodologies and the limitations and challenges inherent in BIVID-MaP into the Discussion section. We believe these revisions significantly strengthen our manuscript and demonstrate the unique value of BIVID-MaP for studying RNA-small molecule interactions.

Below, we provide detailed responses to each of this reviewer's comments.

#Reviewer 1 - Major Comment 1

Clarifying the Significance Relative to Existing Methods

As the authors themselves mention, several alternative techniques exist for mapping RNA-small molecule interactions, such as Chem-CLIP [J. Am. Chem. Soc. 144, 22 11620–11625 (2022)] and mutational profiling methods like SHAPE-MaP [Methods 167, 105–116 (2019)] and DMS-MaPseq [Nat. Methods 14, 75–82 (2017)].

While the authors briefly mention alternative approaches, a more detailed discussion of how BIVID-MaP complements or outperforms existing methods (e.g., Chem-CLIP, SHAPE-MaP) would help readers appreciate the unique value of this method. A direct comparison using the same RNA–ligand system could provide especially compelling support. For example, can BIVID-MaP identify RNA interaction sites with berberine that are not detectable by SHAPE-MaP or DMS-MaPseq? Or does it offer improved binding site resolution?

Response 1:

We thank the reviewer for this valuable suggestion. Following this recommendation, we additionally performed both SHAPE-MaP and DMS-MaPseq experiments in the presence of RNA-binding small molecules (**new Extended Data Fig. 5 and Supplementary Figs. 5–8**). Our comparative analysis

demonstrates that BIVID-MaP exhibits superior sensitivity and specificity compared to both conventional mutational profiling methods.

Key findings from the comparative analysis:

- **SHAPE-MaP:** This method rarely detected berberine–RNA G4 interactions (**Extended Data Fig. 5, Supplementary Figs. 5, 6**).
- **DMS-MaPseq:** While this method identified binding to a subset of G4 structures, it failed to detect bindings that BIVID-MaP successfully identified (**Extended Data Fig. 5, Supplementary Figs. 5, 6**).
- **Non-G4 structure detection:** BIVID-MaP successfully captured the SMN-C2 binding to GA-rich hairpin loops (**Fig. 1e–f, Supplementary Fig. 2**), whereas both SHAPE-MaP and DMS-MaPseq failed to detect this interaction (**Supplementary Fig. 7c, d**).

Given the lower sensitivity and specificity of SHAPE-MaP and DMS-MaPseq, BIVID-MaP is better suited for stringent detection of SNV-specific binding, which is a primary objective of this study. **Indeed, BIVID-MaP was the only method that successfully detected SNV-specific binding in the CD44 G4 I8 variant (Fig. 2d, Supplementary Fig. 8).** We assume the reason can be explained partially by the modification sites of reagents and base preferences, and have added the text below (**Supplementary Discussion**).

Changes made to the manuscript:

- Added new comparative data in **Extended Data Fig. 5** and **Supplementary Fig. 5–8**
- Added new text in the Results section, 8th paragraph: *"To evaluate the performance of BIVID-MaP relative to other mutational profiling methods, we conducted SHAPE-MaP^{7, 22, 23} and DMS-MaPseq⁸ in the presence and absence of RNA-binding ligands to compare mutation rates (Extended Data Fig. 5a). SHAPE-MaP failed to detect RNA-small molecule interactions identified by BIVID-MaP, as no significant differential RT-mutations were observed in positive controls compared to negative controls (Extended Data Fig. 5b, Supplementary Fig. 5, Supplementary Fig. 6, Supplementary Fig. 7). DMS-MaPseq missed some berberine-G4 structure interactions (G4 pre-mir-1229, G4 BCL2, G4 (UGGU)₆), which are clearly detected by BIVID-MaP (Extended Data Fig. 5b, Supplementary Fig. 5). Furthermore, DMS-MaPseq could not detect SMN-C2 binding to GA-rich hairpin loops (Supplementary Fig. 7c, d). These results demonstrate that BIVID-MaP provides superior sensitivity and specificity for detecting RNA-small molecule interactions in our validation system."*
- Added new text in the Results section, 10th paragraph: *"Indeed, BIVID-MaP was the only method among those evaluated that identified this SNV-specific binding, which was not replicated by either SHAPE-MaP or DMS-MaPseq (Supplementary Fig. 8)."*
- Added new text in the Supplementary Discussion section entitled **"The mechanistic differences between mutational profiling approaches can implicate RNA-small molecule interaction detection"**: *"The limited detection capability of conventional methods can be partially explained by their modification preferences. DMS predominantly modifies the nucleobases of cytosine and adenine, while SHAPE-MaP modifies the phosphate backbone. However, direct ligand binding does not always occur at these specific sites with high specificity, which may explain why these methods fail to consistently detect RNA-small molecule interactions."*

#Reviewer 1 - Major Comment 2

2. Generality Across RNA Structures

In this study, the authors synthesized and evaluated VQ-conjugated berberine, CMA, and SMN-C2. However, Figures 2 and subsequent figures primarily focus on berberine binding to RNA G-quadruplex structures.

Although Figures 2–4 show binding site identification for berberine–G-quadruplex interactions, the broader applicability of this method to other RNA structural motifs remains unclear.

VQ's covalent reactivity is known to exhibit nucleobase (e.g., toward thymine-N3, NAR 2019, 47, 6578) selectivity. Furthermore, it is possible that the reactivity varies depending on the angle of the π^* orbital. Therefore, its labeling efficiency may not be uniform across diverse RNA structural contexts. In contrast, photoaffinity labeling with diazirines seems to offer less biased reactivity, which may lead to more comprehensive mapping.

Demonstrating that VQ can form covalent bonds at multiple binding sites, regardless of nearby base composition and RNA structures (at least some G-quadruplex structures or sequences), would support its broader utility. Does the flexibility of the linker contribute to reducing selectivity?

Response 2:

We appreciate the reviewer's important question regarding the applicability of BIVID-MaP to non-G4 structures and the concern about VQ's nucleobase selectivity.

Applicability of this method to other RNA structural motifs:

We acknowledge that our study has primarily focused on variant-specific binding detection in RNA G-quadruplex structures. **To reflect this more accurately, we have revised the Abstract and Introduction to specify the focus on G-quadruplex structure related to discovery of variant-specific interactions, adjusting according to the scope of our claims.**

We would also like to clarify that our original data already demonstrated that BIVID-MaP can detect the binding to non-G4 structures, although this was not sufficiently described in the manuscript. To clearly illustrate this capability, we have added additional data showing selective interaction with the hairpin loop structure of our method.

- **SMN-C2 binding to GA-rich hairpin loop structures:** As presented in **Fig. 1c**, SMN-C2, an RNA binder identified during branaplam development, is known to bind GA-rich hairpin loops (Tang Z. et al., 2021, NAR). Notably, the increased RT-deletion was not observed in the negative control, which contained the same GA-rich sequence in a stem structure (**Supplementary Fig. 2**). Our BIVID-MaP data successfully captured this non-G4 structure selectivity.

Addressing base selectivity of VQ:

We acknowledge the reviewer's concern regarding VQ's selectivity toward specific nucleobases and potential variations in labeling efficiency depending on the π^* orbital angle. Indeed, our data confirm that VQ exhibits bias toward uracil residues. In G4 sequences completely lacking uracil, we were unable to detect the binding (G4 pre-mir-6850 in **Supplementary Figs. 4, 5**). However, conversely, we identified cases in which sequences lacking uridines within the G4 structure itself could generate detectable binding signals through modification of uridine structurally close G4 motif (**as shown in the Figure below**). This suggests that binding can be captured even in the absence of uracil residues within the target structure, as long as reactive bases are present in close proximity to the G4 motif. This partially mitigates the limitation imposed by the uracil selectivity of VQ. **Accordingly, we have revised the Discussion section to include these points.**

Figure | Uridine structurally close to the G4 structure exhibits an RT-deletion signature.

(a) RT-deletion profile showing the interaction between Berberine-VQ-N₃ (NMe₂) and PA2G4 5' UTR. The bases showing the significant RT-deletions (33U, 37U) are highlighted in yellow. (+) indicates guanine residues forming a G-quadruplex in the predicted RNA secondary structure using RNAfold. **(b)** The predicted secondary structure of PA2G4 5' UTR. The G4 structure region is constrained as an unpaired loop and is highlighted in purple. The bases showing the significant RT-deletions is distant from predicted G4 forming region in sequence, but its structurally close. This indicates that the binding can be detected through uridine at structurally close to target RNA structure, even when uridine is not present within the target RNA structure.

The flexibility of the linker can reduce selectivity

As the reviewer mentioned, the flexible linker enables modification of more distal bases, thereby reducing positional bias due to VQ selectivity. Our preliminary validation confirms the importance of linker optimization for achieving less selective binding detection. Specifically, a more flexible linker improved the modification efficiency of SMN-C2 (as shown in the Figure below). We have added new text in the Methods section to inform the importance of linker flexibility to BIVID-MaP users.

Figure | Linker flexibility affects the detection sensitivity of VQ-conjugated SMN-C2.

(a–b) Chemical structure of VQ-conjugated SMN-C2 with a long linker **(a)** or a short linker **(b)**.

(c) Gel shift assay showing the different modification efficiency between SMN-C2-VQ with different linker length. From bottom to top, the gel shift bands show unmodified RNA, a 1:1 covalent complex (one modifier per RNA) and a 1:2 covalent complex (two modifiers per RNA). SMN-C2-VQ with a long linker exhibited more efficient modification than that with a short linker. This indicates that linker flexibility affecting the efficiency of binding detection by BIVID-MaP.

Rationale for choosing VQ over photoaffinity labels:

The reviewer mentions that photoaffinity labeling with diazirines might offer less biased reactivity. While diazirine-based UV crosslinking is widely used and can react with nearby C–H, N–H and O–H bonds (Brunner J et al., 1980, *J Biol Chem*, *Dubinsky L et al.*, 2012, *Bioorg Med Chem*), this non-selective reactivity results in heterogeneous modification sites, potentially targeting sugar moieties or various positions on nucleobases. This heterogeneity could lead to inconsistent reverse transcription signatures, complicating data interpretation.

In contrast, VQ provides uniform modification at the nucleobase level (specifically uracil-N3), which generates consistent deletion signatures during reverse transcription. Since our primary objective was to detect variant-specific binding through deletion-based mutational profiling, we chose VQ for its ability to produce uniform, predictable modification structures that yield robust deletion signals. Previous research on frameshift deletions induced by DNA polymerase suggests that planar modifications to the base can induce deletions by providing a stacking surface (Zhang H et al., 2009, *J Biol Chem*). We expected that this mechanism is also involved in the homogeneous reverse transcription deletion signature observed with VQ, as supported by our molecular modeling and MD simulation analyses demonstrating VQ's stacking ability (new **Supplementary Fig. 1**). The future development of novel modifiers that can modify bases other than uracil and provide an appropriate stacking surface is expected to enable the bias-free detection.

We have now added new text and data in the Results section and Discussion section to properly show this design consideration and the potential for future methodological advancements of VQ, which was not sufficient in previous manuscript.

Changes made to the manuscript:

- Added experimental data for the detection of loop-specific binding of SMN-C2 (**Supplementary Fig. 2**).

- Added molecular dynamics simulation data showing that VQ provides a stacking surface to the complementary cDNA base. (**Supplementary Fig. 1**)
- Changed the text in the Abstract to clarify that our discovery of variant-specific RNA structure-small molecule interactions was limited in G4 structures: *“Using BIVID-MaP, we uncovered numerous variant-specific interactions between a G-quadruplex (G4) binding small molecule and RNAs harboring single-nucleotide variants. Several cancer-associated somatic mutations significantly influence the binding intensity of a small molecule by affecting target G4 structures”*
- Changed the text in the Introduction to clarify that our discovery of variant-specific RNA structure-small molecule interactions was limited in G4 structures: *“Using this approach, BIVID-MaP discovered numerous cancer-associated somatic mutation-specific interactions between 5' UTRs and a G-quadruplex (G4)-binding small molecule.”*
- Added new text in the Results section, 5th paragraph to clarify that BIVID-MaP itself can be applied to non-G4 structures: *“Furthermore, to demonstrate whether our method can detect binding to different types of RNA structures, we tested SMN-C2, which is an analog of an FDA-approved RNA-targeting drug for spinal muscular atrophy²⁰. SMN-C2-VQ (NMe₂) treatment resulted in significant RT deletions in the target hairpin structure in Loop GAAGGAAGG²¹ (Fig. 1c-f). Notably, the significant RT deletions were diminished when target purine-rich sequences were in the stem structure, meaning that our method recognizes the selective binding to the loop structure of SMN-C2 (Supplementary Fig. 2). These results demonstrate that the VQ-conjugated RNA-binding moiety can be replaced with other RNA-binding compounds.”*
- Added new text in the Methods section entitled "Linker design and probe synthesis" to explain that linker flexibility is important for less biased detection: *“The optimal linker length is crucial for efficient and unbiased detection, as linkers that are too short may result in insufficient modification yield.”*
- Added new text in the Results section, the 1st to 3rd paragraph to clarify why VQ was used in our method: *“Previous research on frameshift deletions induced by DNA polymerase indicates that planar modifications, such as 1, N²-etheno-dG, induce deletions by providing a larger stacking surface than canonical bases⁹. The π - π stacking interactions of the modified base with the terminal base of the elongating strand and the incoming dNTP are considered crucial for this process. Based on this mechanism, we hypothesized that a similar modification on RNA would induce RT deletions. “, “We previously reported a modifier comprising an RNA-binding small molecule (Rbs) conjugated to a dimethylamino-protected vinyl-quinazolinone, termed VQ (NMe₂)^{10, 11}. This Rbs-VQ (NMe₂) can be activated to the vinyl VQ form, enabling covalent bonding with bases, especially uridine bases (U), near a target RNA structure. The N3 position of U is the primary modification site. We investigated whether this VQ-modified U could induce deletions near the Rbs binding sites.”, “Our molecular modeling and MD simulation analyses of an RNA/cDNA complex containing a VQ-modified base suggest that the VQ moiety stacks with the terminal base of the growing cDNA strand (at the -1 position), thereby occupying the incoming nucleotide binding position (Supplementary Fig. 1a, b). This stacking persisted in the complex with Thermostable Group II Intron Reverse Transcriptase (TGIRT)¹², a commonly employed RT enzyme in conventional mutation profiling⁸ (Supplementary Fig. 1c). We also observed a potential pocket for accommodating the Rbs and linker regions of the VQ modifier. To develop BIVID-MaP for detecting variant-specific interactions of small molecules, we first assessed how VQ-mediated covalent modification induces the three main types of RT mutations (i.e., substitution, deletion, and insertion).“*
- Added new text in Discussion section, 6th paragraph to mention about the limitation of VQ and future optimization of the modifying moiety : *“Regarding RNA-modifying moieties, there remains scope for optimization. As previously reported, VQ exhibits high selectivity towards U bases. Consequently, this method missed the berberine-G4 interactions when sequences completely lack U (G4 pre-mir-6850, Supplementary Figs. 4, 5). Conversely, even when U is absent within*

*the target RNA structure itself, structurally proximate U residues outside the motif may still serve as reaction sites, which could partially mitigate this bias. From this perspective, the Rbs-VQ linker requires appropriate length and flexibility to enable modification of structurally adjacent U residues. Furthermore, prior studies on frameshift deletions⁹ and our MD simulations support VQ as one of the suitable moieties that provides a stacking surface and can efficiently induce RT deletion (**Supplementary Fig. 1**). Future work introducing novel modifiers with sufficient stacking ability while targeting other bases could extend this approach into a less biased platform.”*

3. Generality Across Small Molecules

This study largely centers on berberine-G4 interactions. As mentioned in Comment 2, the reactivity of VQ seems to be more selective than that of photoaffinity labels.

To support generalizability, the authors should investigate interactions between RNA and additional small molecules (e.g., CMA or SMN-C2) and compare the results with those obtained from other profiling methods, such as SHAPE-MaP or DMS-MaPseq.

If further validation with other small molecules is not currently feasible, the abstract, introduction, discussion, and conclusion should be narrowed to focus specifically on the berberine–G-quadruplex or G-quadruplex. Alternatively, using therapeutically more relevant RNA-binding lead compounds could clarify the significance of compound-specific profiling, even if generalizability is lower.

Response 3:

We thank the reviewer for this important comment.

Following the reviewer's suggestion, we performed SHAPE-MaP and DMS-MaPseq experiments to demonstrate BIVID-MaP's generalizability across different compound classes (**Supplementary Fig. 7**). These new comparative analyses show that, similar to the results observed with berberine (Response 2), BIVID-MaP successfully detects SMN-C2 binding to GA-rich hairpin loops with superior sensitivity compared to conventional methods. Notably, both SHAPE-MaP and DMS-MaPseq failed to detect SMN-C2–RNA interactions while BIVID-MaP successfully captured these bindings (**Supplementary Fig. 7**). These results demonstrate that BIVID-MaP's applicability is not limited to G-quadruplex binders.

Additionally, we clarified the therapeutic value of SMN-C2 for spinal muscular atrophy, as our previous explanation of its therapeutic significance was insufficient.

Changes made to the manuscript:

- Added comparative experimental data with SHAPE-MaP and DMS-MaPseq for CMA and SMN-C2 (**Supplementary Fig. 7**).
- Clarified SMN-C2's identity as a branaplam analog and its therapeutic relevance and emphasized that SMN-C2 binds GA-rich hairpin loops (non-G4 structures) in the Results section, 5th Paragraph. *“Furthermore, to demonstrate whether our method can detect binding to different types of RNA structures, we tested SMN-C2, which is an analog of an FDA-approved RNA-targeting drug for spinal muscular atrophy²⁰.”*
- Demonstrated BIVID-MaP's superior performance across different compound classes and RNA structural motifs in Results section, 8th paragraph: *“SHAPE-MaP failed to detect RNA-small molecule interactions identified by BIVID-MaP, as no significant differential RT-mutations were observed in positive controls compared to negative controls (**Extended Data Fig. 5b, Supplementary Figs. 5–7**).”, “Furthermore, DMS-MaPseq could not detect SMN-C2 binding to GA-rich hairpin loops (**Supplementary Fig. 7c, d**). These results demonstrate that BIVID-MaP provides superior sensitivity and specificity for detecting RNA-small molecule interactions in our validation system.”*

4. [Supplementary] Potential for Cellular Application

This is a supplementary comment and does not require additional experiments.

SHAPE-MaP and DMS-MaPseq can be performed in living cells. If BIVID-MaP could also be adapted for in vivo use, it would significantly enhance its relevance to drug discovery.

However, the membrane permeability of VQ-conjugated compounds may be a concern. A discussion of this point in the manuscript would help readers better understand the potential and limitations of this method.

Response 4:

We thank the reviewer for this insightful comment regarding the potential for cellular application of BIVID-MaP.

While the reviewer noted that additional experiments were not required, we recognized that addressing the practical feasibility of BIVID-MaP in cellular environments would strengthen the manuscript, particularly as other reviewers also raised this concern. Accordingly, although we did not perform experiments in live cells, **we conducted additional assays under cell lysate to mimic cellular conditions** (e.g., in the presence of proteins and relevant cellular factors) to evaluate BIVID-MaP's performance. We optimized the reaction conditions to shorten the modification reaction and successfully performed BIVID-MaP in the presence of cell lysate (**Extended Data Fig. 9**).

As the reviewer correctly points out, membrane permeability of VQ-conjugated compounds may present a challenge for direct in vivo applications. Detection of RNA-small molecule interactions in the complex intracellular environment will require further compound optimization, potentially including cell-permeable VQ derivatives. **We clarified in the Discussion section that further optimization including the improvement of membrane permeability is required for intracellular application.**

Changes made to the manuscript:

- Added new data showing BIVID-MaP performance in cell lysate conditions (**Extended Data Fig. 9**)
- Added new text in the Discussion section, 8th paragraph to show the cell lysate application and future potential for intracellular applications: *“Considering the broader application of BIVID-MaP, fast-reacting modification chemistry is well-suited to the dynamic and unstable RNA structural environment in cells⁴⁵⁻⁴⁷, where G4 structures, in particular, are often unfolded under complex cellular contexts⁴⁸. Importantly, we demonstrated that pre-activation of the VQ-precursor by a 4-hour incubation with methyl vinyl ketone (MVK) enables rapid modification, even when the reaction time is shortened to 10 minutes (**Extended Data Fig. 9a, b**). This pre-activation allowed us to detect berberine-G4 interactions in HeLa cell lysates with sensitivity similar to that observed in the buffer, confirming that a 10-minute incubation is sufficient for detecting berberine binding (**Extended Data Fig. 9c-e**). A possible concern is that glutathione in the cellular environment might react with VQ, which could suppress its modification efficiency. In our reaction system, MVK not only accelerates the reaction but also minimizes undesired reactions with intracellular thiols by acting as a trapping reagent. This enables efficient modification even under conditions that mimic the complex cellular environment, thus suggesting the feasibility of extending BIVID-MaP to intracellular applications, although further optimization of the VQ-conjugated compounds, including improvements in membrane permeability, is still needed.”*

[Minor Comments]

1. P.8, Lines 18–20 et al.:

As discussed, this method may enable the identification of structural changes in RNA resulting from single-base mutations at a level that structural prediction tools cannot predict. However, compounds like berberine often induce and stabilize specific RNA conformations upon binding.

Therefore, any observed changes in binding patterns should not be attributed solely to structural alterations of RNA in the absence of the compound. The authors should clearly state that such changes may result from structure induction upon small-molecule binding.

Response 5:

We thank the reviewer for this important clarification. We agree with the reviewer's opinion that small molecules such as berberine can induce and stabilize specific RNA conformations upon binding, such that observed changes in binding patterns may reflect not only pre-existing structural differences but also differences in compound-induced structural changes.

Accordingly, we have revised the Discussion to clearly state that SNV-dependent binding differences detected by BIVID-MaP may result from both intrinsic structural alterations and structure induction upon small-molecule binding.

Changes made to the manuscript:

Added new text in the Discussion section, 9th paragraph: *“Importantly, the SNV-dependent binding differences detected by BIVID-MaP may result from both intrinsic structural variations and differential structure induction by ligand binding. Small molecules like berberine can induce and stabilize specific RNA conformations^{11, 49, 50} and single nucleotide changes may alter the propensity for such compound-induced structural transitions. Therefore, the observed changes in binding patterns reflect not only pre-existing structural differences but also conformational changes caused by small-molecule recognition.”*

2. Extended Data Figure 1 (Berberine/TGIRT):

The positions of observed deletions and substitutions differ, with substitutions appearing more widespread.

The manuscript should clarify the interpretation of these patterns.

If deletions are prioritized as binding indicators while substitutions are ignored, the authors should provide a rationale to justify this analytical focus.

Response 6:

We thank the reviewer for this important question regarding the interpretation of deletion and substitution patterns. We have two key reasons for prioritizing deletion rather than substitution, which were not sufficiently explained in the previous manuscript.

1. **Deletions are more sensitive than substitutions for detecting binding sites.** Specifically, when evaluating berberine-G4 interactions, we frequently observed false-positive signals with substitutions (**Supplementary Fig. 16**). Such false positives were not due to binding, but instead to RT-substitutions occurring in the region with repetitive nucleotides. In contrast, these non-specific signals were not observed in the deletion analysis, demonstrating that deletions provide specific detection of genuine binding sites. This finding was a key factor in our decision to prioritize deletions for BIVID-MaP analysis.
2. **Deletion-based mutational profiling is more effective for the detection of SNV-specific binding, which is our primary objective.** Recognizing substitutions as binding sites carries the

risk of misinterpreting the target SNV with the binding itself. In contrast, deletions allow for a clear distinction between the target SNV and the binding site.

For these reasons, we prioritized deletions as the primary mutational signature in BIVID-MaP analysis, as they offer both higher sensitivity and better positional accuracy for identifying variant-specific RNA-small molecule interactions, which is essential for our study objectives.

We have now clarified this rationale in the manuscript.

Changes made to the manuscript:

Added new text in the Discussion section, 3rd paragraph: *“For two key reasons, BIVID-MaP employs RT deletions instead of RT substitutions. First, RT deletions are more sensitive and specific than RT substitutions for detecting binding sites. In our berberine-G4 interaction detection, RT substitutions frequently produced false-positive signals in repetitive nucleotide regions (**Supplementary Fig. 16**), whereas such non-specific signals were not observed in deletion-based analysis. Second, detecting nucleotide substitutions carries a risk of misinterpreting the variant of interest as mutations induced by chemical modifications. RT deletions, however, clearly distinguish single-nucleotide substitution variants. Therefore, BIVID-MaP could provide a more accurate means to detect binding specific to single-nucleotide substitution variants.”*

3. Extended Data Figure 1 (CMA and SMN-C2/TGIRT):
Substitutions are also observed with CMA and SMN-C2. However, their impact on analysis is not discussed.
The authors should clarify whether these substitutions could lead to false positives or otherwise affect interpretation.

Response 7:

We thank the reviewer for this comment.

As discussed in our response to Minor Comment 2, substitutions lack the sensitivity required for accurate binding site identification and could produce false-positive signals (**Supplementary Fig. 16**). Because these false signals are TGIRT-dependent rather than compound-dependent, this problem is consistent across all compounds tested, including CMA and SMN-C2, as the reviewer suggested.

Therefore, we prioritize deletions as the primary mutational signature for all compounds to ensure robust detection of variant-specific interactions while minimizing false positives. We applied this analytical approach uniformly throughout our study.

Reviewer #2

In this study, Miyashita et al. developed BIVID-MaP, a high-throughput method for detecting RNA-small molecule interactions that combines binding-dependent covalent modification with profiling of deletions upon reverse transcription via deep sequencing. From the data, they revealed numerous variant-specific interactions between an RNA-binding small molecule and RNA harboring single-nucleotide variants. In addition, several cancer-associated somatic mutations significantly influence the binding intensity of a small molecule by affecting target RNA structures. The overall data may contribute to the development of RNA-targeting drugs in the future.

Overall, this work potentially fits the aim and scope of Nature Communications. The authors have tried to address a research gap that is often gone unnoticed and have come up with a novel and practical solution to understand/report it. There are also individual examples and significance shown in this work, as they have demonstrated that it is linked to cancer-associated somatic mutations. However, there are some additional experiments and controls that are needed to support the claims of this manuscript, as well as to enhance the quality of data and the impact/implication of the study. The author also needs to perform some of these experiments in cellular context/lysate to showcase the usefulness of the method, as RNA structure in vitro and in cells can be very different, and the presence of other components may affect the chemical binding and crosslinking efficiency. I hope the authors will make an effort to revise the manuscript substantially and accordingly, which is required to meet/exceed the high standard of this prestigious journal. Once the authors can fully address the major and minor comments, the scientific quality of the manuscript will be substantially improved and I will be happy to re-review it.

We sincerely thank the reviewer for the thoughtful and constructive evaluation of our manuscript. We greatly appreciate the recognition of our work's novelty and its potential contribution to RNA-targeting drug development.

We have carefully considered all of this reviewer's comments and have conducted substantial additional experiments and revisions to address the raised concerns. Most notably, in response to the reviewer's critical point regarding the applicability of BIVID-MaP in cellular context/lysate, **we have performed new experiments using cell lysate conditions and developed an optimized protocol with significantly reduced reaction time**. These experiments demonstrate that BIVID-MaP functions effectively in the presence of cellular proteins and other components. **By dramatically shortening the reaction time from 8 hours to 10 minutes**, we have minimized potential interference from cellular factors while maintaining high sensitivity and specificity for detecting RNA-small molecule interactions.

Furthermore, following the reviewer's suggestion, **we have added biochemical experiments and analyses to appropriately discuss the formation of the RNA structure**.

Below, we provide detailed point-by-point responses to each comment.

Major comments:

1. Figure 1. It seems that the small molecule will mostly interact with U nearby the target binding region. This will lower the chance of crosslinking. Did they try other crosslinking groups or consider other strategies?

Response 1:

We appreciate the reviewer's valuable comment regarding the important aspect of base selectivity. Our current study focused specifically on VQ-based modifiers, and therefore we did not experimentally evaluate other crosslinking chemistries. Below, we provide our rationale for this choice and explain how we addressed the uracil selectivity of VQ.

Addressing base selectivity of VQ:

It is acknowledged that our modified moiety primarily interacts with Uracil (U), resulting in a low probability of crosslinking. For instance, the berberine-G4 interaction was not observed in sequences entirely lacking U (G4 pre-mir-6850 in **Supplementary Figs. 4–5**). Conversely, we identified cases in which sequences lacking uridines within the G4 structure itself could generate detectable binding signals through modification of uridine structurally close G4 motif (**as shown in the Figure below**). This suggests that binding can be captured even in the absence of U residues within the target structure, as long as reactive bases are present in close proximity to the G4 motif. This partially mitigates the limitation imposed by the uracil selectivity of VQ. **Accordingly, we have revised the Discussion section to include these points.**

Figure | Uridine structurally close to the G4 structure exhibits an RT-deletion signature.

(a) RT-deletion profile showing the interaction between Berberine-VQ-N₃ (NMe₂) and PA2G4 5' UTR. The bases showing the significant RT-deletions (33U, 37U) are highlighted in yellow. (+) indicates guanine residues forming a G-quadruplex in the predicted RNA secondary structure using RNAfold. **(b)** The predicted secondary structure of PA2G4 5' UTR. The G4 structure region is constrained as an unpaired loop and is highlighted in purple. The bases showing the significant RT-deletions is distant from predicted G4 forming region in sequence, but its structurally close. This indicates that the binding can be detected through uridine at structurally close to target RNA structure, even when uridine is not present within the target RNA structure.

Rationale for choosing VQ over photoaffinity labels:

We selected VQ as the modifying moiety to enable reliable detection of variant-specific binding via deletion-based mutational profiling. Unlike unselectively reactive UV crosslinkers such as diazirines, which generate heterogeneous adducts on sugars and nucleobases (Brunner J et al., 1980, *J Biol Chem*; Dubinsky L et al., 2012, *Bioorg Med Chem*) and have the risk of complicated RT-based readouts, VQ predominantly modifies the N3 position of uridine, which generates a consistent deletion signature during reverse transcription. This predictable behavior is well suited for robust interpretation of variant-dependent binding events. Consistent with prior reports that planar base modifications can induce polymerase frameshift deletions (Zhang H et al., 2009, *J Biol Chem*), we propose that VQ promotes RT deletions by providing a stacking surface. This hypothesis is supported by our molecular modeling and MD simulations (**new Supplementary Fig. 1**), which also suggest VQ is one of suitable modifying moieties for deletion-based mutational profiling .

However, as the reviewer appropriately noted, an alternative modifying module might be more suitable for our methodology. For example, the future development of novel modifiers capable of modifying bases other than uracil while providing an appropriate stacking surface is expected to enable bias-free detection. **To properly inform readers about the current limitations of VQ and the potential for future methodological advancements, we have incorporated this discussion into the manuscript.** We appreciate the reviewer's comment for highlighting a critical aspect of our methodology.

Changes made to the manuscript:

- Added molecular dynamics simulation data showing that VQ provides a stacking surface to the complementary cDNA base. (**Supplementary Fig. 1**)
- Added new text in the Results section, the 1st to 3rd paragraph to clarify why VQ was used in our method: *“Previous research on frameshift deletions induced by DNA polymerase indicates that planar modifications, such as 1, N2-etheno-dG, induce deletions by providing a larger stacking surface than canonical bases⁹. The π - π stacking interactions of the modified base with the terminal base of the elongating strand and the incoming dNTP are considered crucial for this process. Based on this mechanism, we hypothesized that a similar modification on RNA would induce RT deletions.”*, *“We previously reported a modifier comprising an RNA-binding small molecule (Rbs) conjugated to a dimethylamino-protected vinyl-quinazolinone, termed VQ (NMe₂)^{10, 11}. This Rbs-VQ (NMe₂) can be activated to the vinyl VQ form, enabling covalent bonding with bases, especially uridine bases (U), near a target RNA structure. The N3 position of U is the primary modification site. We investigated whether this VQ-modified U could induce deletions near the Rbs binding sites.”*, *“Our molecular modeling and MD simulation analyses of an RNA/cDNA complex containing a VQ-modified base suggest that the VQ moiety stacks with the terminal base of the growing cDNA strand (at the -1 position), thereby occupying the incoming nucleotide binding position (**Supplementary Fig. 1a, b**). This stacking persisted in the complex with Thermostable Group II Intron Reverse Transcriptase (TGIRT)¹², a commonly employed RT enzyme in conventional mutation profiling⁸ (**Supplementary Fig. 1c**). We also observed a potential pocket for accommodating the Rbs and linker regions of the VQ modifier. To develop BIVID-MaP for detecting variant-specific interactions of small molecules, we first assessed how VQ-mediated covalent modification induces the three main types of RT mutations (i.e., substitution, deletion, and insertion).”*
- Added new text in Discussion section, 6th paragraph to mention about the limitation of VQ and future optimization of the modifying moiety : *“Regarding RNA-modifying moieties, there remains scope for optimization. As previously reported, VQ exhibits high selectivity towards U bases. Consequently, this method missed the berberine-G4 interactions when sequences completely lack U (G4 pre-mir-6850, **Supplementary Figs. 4, 5**). Conversely, even when U is absent within the target RNA structure itself, structurally proximate U residues outside the motif may still serve as reaction sites, which could partially mitigate this bias. From this perspective, the Rbs-VQ linker requires appropriate length and flexibility to enable modification of structurally adjacent U residues. Furthermore, prior studies on frameshift deletions⁹ and our MD simulations support VQ as one of the suitable moieties that provides a stacking surface and can efficiently induce RT deletion (**Supplementary Fig. 1**). Future work introducing novel modifiers with sufficient stacking ability while targeting other bases could extend this approach into a less biased platform.”*

2. Figure 1. In addition, the crosslinking time is very slow, taking 8 or more hours. As RNA is quite unstable in vitro and in cells, and these chemicals may also be somewhat toxic. Has the authors demonstrated the dosage and time-dependent treatment on RNA degradation and cell viability, as well as other off-target binding and crosslinking in a cellular context/lysate?

Response 2:

We thank the reviewer for this critical comment regarding reaction time and applicability to cellular contexts. We agree that the long reaction time (8+ hours) could be a limitation, particularly given concerns regarding RNA instability and potential toxicity concerns. To address this, **we have developed a new activation method and conducted new experiments in cell lysate to demonstrate BIVID-MaP's applicability in more biologically relevant conditions.**

Development of a new activation method:

To overcome the limitation of long reaction times, we developed a novel strategy involving the pre-incubation of a VQ-conjugated compound. This approach enables the pre-activation of VQ before the alkylation step by utilizing methyl vinyl ketone (MVK) (**Extended Data Fig. 9a**). MVK not only accelerates the reaction but also inhibits competitive reactions with intracellular thiols by acting as a trapping reagent. Consequently, this allows us to skip the elimination reaction, which was previously the rate-determining step, and significantly accelerates the reaction (**Extended Data Fig. 9b**).

Demonstration in cell lysate:

We next performed model experiments to evaluate BIVID-MaP in cellular environments using HeLa cell lysate (**Extended Data Fig. 9c**). The results demonstrated that the modification efficiency was achieved within the lysate environment over a 10-60 minute reaction time (**Extended Data Fig. 9d**). Significantly, **BIVID-MaP detects berberine-G4 interactions with only a 10-minute reaction time**, in the presence of cellular components (**Extended Data Fig. 9e**). This is comparable to the reaction times typically employed in conventional mutational profiling and crosslinking-based binding detection methods (Zeller, M. J. et al., 2022, PNAS, Fang, L. et al., 2023, Nat. Chem.).

This new method directly addresses concerns about RNA instability and demonstrates that BIVID-MaP can function in complex biological environments containing proteins and other cellular factors. Detection of RNA-small molecule interactions in the complex intracellular environment will require further compound optimization, potentially including cell-permeable VQ derivatives. **We clarified in the Discussion section that further optimization is required for intracellular application.**

Despite these limitations, **our newly developed method demonstrates that BIVID-MaP can function effectively in cell lysate with dramatically reduced reaction times**, addressing the major concerns about RNA stability and practicality for biological applications.

Changes made to the manuscript:

- Added new data showing the rapid modification through the pre-incubation method with time-dependent validation (10, 30, 60 min) (**Extended Data Fig. 9a, b**)
- Added new data showing BIVID-MaP performance in cell lysate conditions (**Extended Data Fig. 9c–e**)
- Described the new catalytic method and its performance and potential concern of VQ for future applications in the Discussion section, 8th paragraph: *“Considering the broader application of BIVID-MaP, fast-reacting modification chemistry is well-suited to the dynamic and unstable RNA structural environment in cells^{45–47}, where G4 structures, in particular, are often unfolded under complex cellular contexts⁴⁸. Importantly, we demonstrated that pre-activation of the VQ-precursor*

by a 4-hour incubation with methyl vinyl ketone (MVK) enables rapid modification, even when the reaction time is shortened to 10 minutes (Extended Data Fig. 9a, b). This pre-activation allowed us to detect berberine-G4 interactions in HeLa cell lysates with sensitivity similar to that observed in the buffer, confirming that a 10-minute incubation is sufficient for detecting berberine binding (Extended Data Fig. 9c–e). A possible concern is that glutathione in the cellular environment might react with VQ, which could suppress its modification efficiency. In our reaction system, MVK not only accelerates the reaction but also minimizes undesired reactions with intracellular thiols by acting as a trapping reagent. This enables efficient modification even under conditions that mimic the complex cellular environment, thus suggesting the feasibility of extending BIVID-MaP to intracellular applications, although further optimization of the VQ-conjugated compounds, including improvements in membrane permeability, is still needed.”

3. Figure 1. It seems the linker is different for each case; however, it is not described in the maintext at all. It is suggested to show the data in results, or at least discuss the design of linker somewhere, as I assume it is also important for the binding and crosslinking efficiency, as it should be flexible enough to find the Us nearby. It is unclear at the moment on the design and whether others have been tested. The linker may also play an important role in cellular application, for solubility and cell permeability later.

Response 3:

We thank the reviewer for this important comment regarding linker design. The reviewer raises an excellent point that linker properties, including length, flexibility, and chemical composition, may also influence solubility and cell permeability, which are important considerations for future cellular applications of BIVID-MaP. **Based on this suggestion, we have detailed the rationale behind our current linker selection below and have incorporated text addressing this important aspect in the revised manuscript.**

Linker optimization studies

We agree with the reviewer's opinion that the linker is a critical parameter for binding and crosslinking efficiency, as it must be sufficiently flexible to allow VQ to reach nearby bases.

We conducted preliminary optimization studies examining different linker lengths and their effects on modification efficiency (**as shown in the Figure below**). **Our results demonstrate that linker length affects the reactivity and detection sensitivity of BIVID-MaP.** Linkers that are too short fail to react with certain RNA targets, confirming that appropriate linker length is essential for optimal performance. These data support the importance of linker flexibility for enabling VQ to reach accessible uracil residues in the vicinity of binding sites, thereby mitigating the positional constraints imposed by VQ's nucleobase selectivity (as also discussed in our response to Reviewer 1, Major Comment 2)

Figure | Linker flexibility affects the detection sensitivity of VQ-conjugated SMN-C2.

(a–b) Chemical structure of VQ-conjugated SMN-C2 with a long linker **(a)** or a short linker **(b)**.

(c) Gel shift assay showing the different modification efficiency between SMN-C2-VQ with different linker length. From bottom to top, the gel shift bands show unmodified RNA, a 1:1 covalent complex (one modifier per RNA) and a 1:2 covalent complex (two modifiers per RNA). SMN-C2-VQ with a long linker exhibited more efficient modification than that with a short linker. This indicates that linker flexibility affecting the efficiency of binding detection by BIVID-MaP.

Rationale for linker selection

For our conjugate design, we selected specific linkers that have been validated in previous publications for their favorable properties. PEG linkers provide several advantages: (1) appropriate length and flexibility to accommodate VQ reactivity with nearby uracils, (2) maintenance of aqueous solubility, and (3) avoidance of aggregation issues that can occur with alkyl (C-C) linkers. The use of previously validated linker structures ensures reliability while facilitating reproducibility of our method (Onizuka K. et al., 2019, *NAR*, Yutong C. et al., *Bioconjugate Chem.*, 2022).

A longer linker was employed in the SMN-C2-VQ when interacting with the hairpin loop structure, in the expectation that it would facilitate access to the distant U base. An alkyl linker was introduced in the area near the binding site, where hydrophobic interactions were expected, which was subsequently connected to the VQ using a PEG linker.

Changes made to the manuscript

- Added explanation of linker selection rationale in new Methods section entitled **Linker design and probe synthesis**: *“VQ-conjugated probes were synthesized as described previously^{10, 11} and in the Supplementary Methods. For linker selection we primarily adopted PEG spacers as we expect PEG linkers to (i) provide sufficient reach and conformational flexibility for the VQ electrophile to access nearby uracils, (ii) maintain aqueous solubility, and (iii) reduce aggregation compared with purely alkyl linkers. In the design of SMN-C2, a longer linker was incorporated to enhance the interaction of SMN-C2-VQ with the hairpin loop structure and facilitate access to the distant U base. Specifically, an alkyl linker was introduced near the binding site to promote hydrophobic interactions, followed by a PEG linker connecting it to the VQ. The optimal linker length is crucial for efficient and unbiased detection, as linkers that are too short may result in insufficient modification yield.”*

4. Figure 1. The % detection seems very low, even though the gel showed ~50% crosslinking. Does it mean the TGIRT is still not doing a very good job in deletion during RT? This also implies that relatively deep sequencing is needed to have a good signal-to-noise level. Can the author also comment on the sequencing requirement for the non-pull down and pull down case? This may also be a limitation of the method.

Response 4:

We thank the reviewer for this important question regarding the relationship between crosslinking efficiency and deletion rates, as well as the sequencing depth requirements for BIVID-MaP.

As described, we observed via gel shift assays that at least over 40% modifications are yielded in positive controls, while the deletion rates in sequencing data are clearly lower. We acknowledge that this discrepancy suggests that not all modification events are efficiently converted into detectable deletion signatures. The exact mechanism and efficiency of this conversion require further investigation and represents an area for potential improvement of the method.

The reviewer is correct that the relatively low deletion rates necessitate adequate sequencing depth to achieve good signal-to-noise ratios. To address this concern and provide concrete guidance for BIVID-MaP applications, **we performed systematic downsampling analysis to determine the minimum sequencing depth required for robust detection**. The result demonstrates that **at least 500 reads per RNA sequence are sufficient** to achieve reliable detection of RNA-small molecule interactions with statistical confidence (New **Supplementary Fig. 3**).

Accordingly, our method is adaptable for standard sequencing methods and can feasibly identify binding sites. This is because the calculated threshold (500 reads) is comparable to the read depth threshold typically used in conventional MaP methods [Siegfried et al., 2014 Nat Methods].

Changes made to the manuscript:

- Added downsampling analysis demonstrating required sequencing depth (**Supplementary Fig. 3**)
- Added new text showing the sequencing requirement for the non-pull down and pull down case in the Results section, 7th paragraph: *“ROC curves, generated from random sampling of sequencing reads, revealed that pull-down samples require only 500 reads per gene for accurate binding detection (**Supplementary Fig. 3a**). This threshold is comparable to those employed by existing mutation profiling methods⁷. In contrast, samples without pull-down failed to reliably distinguish binding even with 5000 reads (**Supplementary Fig. 3b**). This finding indicates that including the pull-down step reduces the necessary sequencing read depth by at least 10-fold.”*

5. Figure 2. The authors should show that the G->A mutation can disrupt the G4 formation by some biophysical/biochemical assays, such as CD, UV melting, NMM staining gel, etc. A comparison to WT is needed. Please also do so for other individual candidates presented in other figures.

Response 5:

We thank the reviewer for raising these important comments regarding the validation of G4 structural formation.

Following the reviewer's suggestion, we performed additional circular dichroism (CD) measurements and an N-methyl mesoporphyrin IX (NMM)-based G4-sensing fluorescence assay on the sequences for

which we had discussed potential G4 involvement (**Fig. 2d, Fig. 3c, Fig. 4f, Supplementary Fig. 15a**). These experiments confirmed, across multiple sequences, G4-structure-dependent changes in binding associated with single-nucleotide variants (SNVs).

The summary is as follows.

- Variant-specific binding in a G4 structure-dependent manner was confirmed as follows. These results were consistent with the differential binding of berberine detected by BIVID-MaP. Importantly, we discovered that both G4 structure-enhancing or destabilizing cancer-associated mutations, which are difficult to predict by in silico RNA structure prediction.
 - CD44 I8 variants
 - CD: **Extended Data Fig. 6a**
 - DAXX variants
 - CD: **Fig. 4b, Supplementary Fig. 12a**
 - NMM: **Fig. 4c, d, Supplementary Fig. 13b**
 - VPS53 variants
 - CD: **Fig. 4h, Supplementary Fig. 12c**
 - NMM: **Fig. 4i, j**
- G4 structure-dependent binding was not clearly confirmed in ING2 variants and WDR11 variants. For these sequences, we have toned down assertions concerning G4 structures or modified our discussion to suggest the possibility of binding to other RNA structures.
 - ING2 variants
 - CD: **Supplementary Fig. 12b**
 - NMM: **Supplementary Fig. 13c**
 - WDR11 variants
 - CD: **Supplementary Fig. 12d**
 - NMM: **Supplementary Fig. 13d**

Changes made to the manuscript:

- Added CD spectrum analysis data in **Fig. 4b, h and Supplementary Fig. 12**
- Added NMM sensing assay data in **Fig. 4c, d, i, j and Supplementary Fig. 13**
- Added new text in the Results section, 15th–16th paragraph: *“To evaluate G4 formation and variant effects in the DAXX 5' UTR, we performed CD spectroscopy and an NMM fluorescence assay, in which NMM shows selective fluorescence enhancement upon binding to G4 structures^{32, 33} (**Fig. 4a**). CD analysis of an extracted G-rich region (1-26nt) confirmed K⁺-dependent G4 folding (**Fig. 4b, Supplementary Fig. 12a**), but did not reveal variant-dependent differences anticipated from berberine-binding, likely because the excised fragment does not fully preserve the original RNA structure.”*, *“We therefore extended the assay to the full-length 5' UTR. The DAXX 5' UTR reference exhibited increased NMM fluorescence in K⁺ relative to Li⁺, indicating the G4 formation in the full-length construct (**Fig. 4c**). Notably, NMM-enhanced fluorescence changed in a variant-specific manner concordant with berberine binding. The 3 A>G significantly increased fluorescence, whereas 18G>A and 20G>A significantly decreased it (**Fig. 4d, Supplementary Fig. 13b**). By contrast, variants outside the G-rich region (73G>A, 84G>U) had no effect. Together, these results indicate that a G4 within the DAXX 5' UTR contributes to berberine interactions and that cancer-associated somatic mutations modulate the formation of the target G4 structure.”*
- Added new text in the Results section, 19th paragraph: *“However, neither the CD spectrum nor NMM fluorescence showed canonical G4 features (**Supplementary Fig. 12b, Supplementary Fig. 13c**). Thus, we could not confirm any clear G4 structure-dependence for the ING2 5' UTR, raising the possibility that G4 formation is too weak to be detected by conventional G4 detection assays, or that these small molecules bind to non-G4 structures.”*

- Added new text in the Results section, 21st paragraph: *"In the WDR11 5' UTR, canonical G4 signatures were not observed by CD spectra nor NMM fluorescence (Supplementary Fig. 12d, Supplementary Fig. 13d), or sequence-based G4 predictors (Supplementary Fig. 11). Additionally, CMA-VQ (NMe₂) also showed an elevated RT-deletion at 52C>G (Supplementary Fig. 10a). One possible explanation for the absence of a clear G4 signature is intercalation of the small molecule into the RNA, as planar cationic aromatics are known to partially intercalate into non-G4 RNAs^{38, 39}. Under this hypothesis, the increased signal is more likely to reflect non-G4 structural reorganization rather than G4 formation"*
- Added new text in the Results section, 23rd paragraph: *"Consistent with this interpretation, both the reference and 78 U>A guanine-rich regions displayed G4 structure-specific CD spectra (Fig. 4h, Supplementary Fig. 12c). Importantly, 78 U>A significantly enhanced NMM fluorescence, confirming increased stabilization of G4 structures by this mutant (Fig. 4i, j)"*

6. Figure 3. For the 2 examples shown, it is hard to understand how the 128 G>A affects the very upstream U. It is better to show the secondary structure or where the G4 structure is.

Response 6:

We thank the reviewer for this helpful suggestion to improve the clarity of Figure 3. The relationship between the 128 G>A mutation and the upstream uracil modification was not clearly illustrated.

The modification of 12U is interpreted to be due to its structurally close to the G4 structure and variant position (128G), despite being distant from them in the sequence, as shown in **Supplementary Fig. 14**.

We clarified the hypothesis regarding the influence of the structure near 128G on upstream U by showing the secondary structures.

Changes made to the manuscript:

- Added new **Supplementary Fig. 14** to show 128G is structurally close to the upstream base showing highest RT-deletion in predicted secondary structure.

7. Figure 3. Did they verify the data with the other G4 ligand or no G4 ligand control (just linker+crosslinking part)? It is important to have these controls.

Response 7:

We thank the reviewer for raising these important points to enhance data verification. Following the reviewer's suggestion, we performed additional BIVID-MaP with the other G4 ligand (**CMA-VQ-N₃**) or no G4 ligand control (**VQ-N₃**, only linker and VQ part) to the RNA sequences related to Figure 3-4. In summary, this experiment further validated that the selective modification observed in BIVID-MaP is directly attributable to the target ligand moiety present in the modification agent.

The summary is as follows.

- **CMA-VQ-N₃ (NMe₂), another G4 ligand (Supplementary Fig. 10a):** CMA-VQ-N₃ (NMe₂) generally showed variant-dependent changes in BIVID-MaP consistent with Berberine-VQ-NMe₂, indicating that these variants can alter the binding of G4-binding small molecules globally. For

VPS53, CMA did not show a clear RT-deletion increase in 78 U>A compared to berberine, suggesting that the mutational effect on G4 structure binding is target ligand-dependent.

- **VQ-N₃, G4 ligand control (Supplementary Fig. 10b)** : The result of VQ-N₃ showed no significant changes with a low-background RT-deletion across almost all tested RNAs, indicating that the observed variant-specific binding is directly dependent on the presence of the target ligand moiety.

Changes made to the manuscript:

- Added new BIVID-MaP data using **CMA-VQ-N₃** and **VQ-N₃** in **Supplementary Fig. 10**.
- Added new text in the Results section, 13th paragraph: *“Additionally, VQ lacking the target small molecule (VQ-N₃) generated only low-level background signals and did not recapitulate the pronounced variant-specific peaks observed with Berberine-VQ-N₃ (NMe₂), supporting that the variant-specific RT-deletions primarily arise from target ligand-dependent binding (Supplementary Fig. 10b).”*
- Added new text in the Results section, 17th paragraph: *“Additionally, to examine effects on other ligands that recognize G4 structures, we performed BIVID-MaP using CMA-VQ-N₃ (NMe₂). Variants in the DAXX 5' UTR reproduced the same RT-deletion signature in 12U observed with Berberine-VQ (NMe₂) (Supplementary Fig. 10a). These results indicate that the cancer-associated somatic variants identified in DAXX can broadly modulate the binding of G4-recognizing ligands by affecting G4 structural formation.”*
- Added new text in the Results section, 19th paragraph: *“Disruption of the G-tract by the 128 G>A variant markedly reduced RT deletions with both Berberine-VQ-N₃ (NMe₂) and CMA-VQ-N₃ (NMe₂), suggesting that this mutation broadly affects ligands that recognize G4 structures (Fig. 3c, Supplementary Fig. 10a).”*
- Added new text in the Results section, 21st paragraph: *“Additionally, CMA-VQ (NMe₂) also showed an elevated RT-deletion at 52C>G (Supplementary Fig. 10a).”*
- Added new text in the Results section, 23rd paragraph: *“Moreover, the absence of a comparable variant-specific increase in the RT-deletion with CMA-VQ-N₃ (NMe₂), relative to Berberine-VQ-N₃ (NMe₂), suggests that the effect of 78U>A may be dependent on the target G4-binding small molecule (Supplementary Fig. 10a).”*

8. Figure 4. The authors may use RNAfold, which has an option to allow prediction of G4. Also, it may be good to put the sequence into some of those G4 prediction software, such as G4NN, G4 hunter. In addition, it seems hard to know where the G4 formation is and where the G4 ligand binding sites are. Again, the data should be verified with other G4 ligand and no G4 ligand control (just linker +crosslinking part)

Response 8):

We thank the reviewer for the valuable suggestion. Following the reviewer's recommendation, we performed additional sequence-based G4 structure predictions, G4NN, G4 Hunter, and cGcC. Importantly, the mutations we identified that caused G4 structure-dependent binding alterations did not show significant changes in any of the G4 prediction scores (**Fig. 4, Supplementary Fig. 11**). This reinforces the finding that our method can detect G4-dependent binding alterations that are difficult to identify using sequence-based prediction alone.

We verified the use of another G4 ligand and a control without any G4 ligand (only the linker and cross-linking part), as detailed in our response to major comment 7.

Changes made to the manuscript:

- Added new BIVID-MaP data in **Supplementary Fig. 11**.
- Added new text in the Results section, 18th paragraph: *“Additionally, sequence-based G4 prediction software (cGcC³⁵, G4Hunter³⁶, G4NN³⁷) showed only modest differences between variants and did not reproduce the pronounced variant-specific changes observed in berberine binding (Supplementary Fig. 11).”*
- Added new text in the Results section, 23rd paragraph: *“Notably, binding was affected in a G4 structure-dependent manner, even though the 78 U>A substitution does not alter the G-tracts that directly influence quadruplex formation¹⁷, and sequence-based G4 prediction remained unchanged (Supplementary Fig. 11).”*

Minor comments:

1. Figure 2. The G4-region should be bolded or underlined for easier identification. The same applies to other figures.

Response:

We thank the reviewer for this helpful suggestion to improve figure clarity.

We have revised Fig. 2 and other relevant figures to clearly highlight the G4 regions by bolding or underlining, making it easier for readers to identify these critical structural elements.

Changes made to the manuscript:

- Revised Fig. 2 and related figures to bold/underline G4 regions for better visualization.
- Applied this improvement consistently across all figures showing RNA sequences with structural motifs.

2. Why do the authors not directly incorporate biotin into the chemical, but use azide? Is it better for synthesis or later cell treatment/application?

Response:

We thank the reviewer for this question regarding our design choice to incorporate azide rather than directly incorporating biotin.

We chose to introduce an azide group because incorporating a bulky moiety could significantly reduce both the alkylation efficiency of VQ and the binding specificity of target small molecules. To minimize these potential risks, we adopted a strategy of introducing the azide group and conjugating the biotin via a click reaction.

3. There are multiple crosslinking bands seen on the gel, and the authors have not explained their origin and whether they will affect the library preparation and analysis.

Response:

We thank the reviewer for this important question regarding the multiple crosslinking bands observed on the gels.

To accurately discuss the origin of multiple crosslinking bands, we performed additional characterization experiments, targeting (UGGU)₆, which exhibited the clearest additional band when modified by Berberine-VQ-N₃ (NMe₂) (**Supplementary Fig. 4**). MALDI-TOF mass spectrometry revealed that not only one alkylation event per RNA molecule but also two alkylation events occurred, suggesting that the additional gel-shift band corresponds to one RNA molecule covalently modified by two Rbs-VQ (**New Supplementary Fig. 18**). **We have clearly indicated the origin of multiple bands in the Results section and in the legends of all figures including gel images.**

These multiple crosslinking events do not negatively affect BIVID-MaP library preparation or analysis. Because **BIVID-MaP relies on deletion signatures during reverse transcription**, each modification site independently generates a deletion signal at its respective position in the sequencing data. Therefore, multiple modification sites simply provide multiple independent readouts of the binding interaction, rather than interfering with detection. It is rather expected to increase the probability of inducing at least one RT-deletion per single RNA molecule, thereby enhancing detection sensitivity. **We have clarified this point in the manuscript to help readers understand that multiple crosslinking events possibly enhance rather than complicate the analysis.**

Changes made to the manuscript:

- Added new MALDI-TOF mass spectrometry data confirming the labeling stoichiometry for representative RNA-probe pairs (1:1 for single-modified RNAs; 1:2 for dual-modified RNAs) (**Supplementary Fig. 18**).
- Added the new text in the figure legend to clarify the labeling stoichiometry for each gel band in **Fig. 1d, Fig. 2e, Extended Data Fig. 9b and Supplementary Fig. 4**.
- **Added the new section in Supplementary Note entitled *MALDI-TOF analysis indicates that the multiple gel-shift bands originate from 1:1 and 1:2 Rbs-VQ:RNA labeling stoichiometries*: “As multiple band shifts were observed in the gel shift assay (Fig. 1d, Fig. 2e, Supplementary Fig. 4), we verified their stoichiometric origin. MALDI-TOF analysis of model G4 RNA (UGGU)₆ revealed that both single and double alkylation events per RNA molecule occurred (Supplementary Fig. 18). This result suggests that the formation of multiple adducts on a single RNA molecule increases the probability of at least one RT deletion per molecule, thereby potentially amplifying the RT-deletion effectively.”**

4. It is not clear to me sometimes the author uses % deletion on the y-axis, and sometimes RT-deletion on the y-axis.

Response:

We thank the reviewer for pointing out this potential source of confusion in our figure labeling. We acknowledge that the term "BIVID signal" could be confusing. To improve clarity, **we have renamed this metric to "RT-deletion (%)"** throughout the manuscript and figures, which more accurately describes the normalization approach and reduces potential confusion for readers.

- Updated all figure labels to use "**RT-deletion (%)**" instead of "BIVID signal"
- Clarified the definitions and calculation methods in the figure legends and Methods section
- Ensured consistent terminology throughout the manuscript

We appreciate the reviewer's attention to this detail, which has helped us improve the clarity of our presentation.

Reviewer #3

Reviewer #3 (Remarks to the Author):

The manuscript by Miyashita et al. describes the application of Binding- and Vinyl-Quinazolinone (BVQ)-based chemical probes previously reported for investigating G-quadruplexes induced or lost in 5' untranslated regions (UTRs) in mRNAs through single-nucleotide variants (SNVs). BVQ probes contain a latent electrophile, that upon binding to an RNA motif-of-interest, are activated for covalent labeling. In this work, BVQ probes are used in concert with a deletion-based mutational profiling method to identify labeling sites in a library of in vitro transcribed 5' UTR sequences from cancer-related genes that contain SNVs. Using a berberine-based probe and the developed "Binding- and Vinyl-Quinazolinone-Induced Deletion-Based Mutational Profiling" (BIVID-MaP) strategy, the team uncovered several SNVs that affect how berberine binds to the RNA (i.e., if a G-quadruplex structures was induced or disrupted due to the mutation). This later aspect is the most exciting part of the work with the potential of uncovering the presence of potentially druggable G-quadruplexes in cancer-relevant RNAs. While the overall concept is exciting for the field, there are several aspects of the manuscript that need improvement before publication.

Response:

We sincerely thank the reviewer for the positive evaluation of the conceptual framework of our work and for recognizing the potential of BIVID-MaP to uncover druggable G-quadruplex structures in cancer-relevant RNAs through SNV-dependent binding profiling.

We have carefully addressed all of this reviewer's comments through substantial revisions and conducted additional experiments to improve the manuscript. Key improvements include:

- **Developed a new catalytic method using MVK** to dramatically reduce reaction time to 10 minutes, addressing concerns about applicability and RNA stability (**Extended Data Fig. 9a, b**)
- **Demonstrated BIVID-MaP functionality in cell lysate**, providing evidence for potential biological applicability in complex cellular environments (**Extended Data Fig. 9c-e**)
- **Provided detailed biochemical characterization of TGIRT's deletion-generating mechanism**, explaining the molecular basis for this unique reactivity with VQ-modified nucleotides (**Supplementary Fig. 1**)
- **Clarified RNA gel visualization methods** and addressed observations of apparent RNA degradation over time
- **Clarified the labeling stoichiometry by additional MALDI-TOF analysis** and the detailed explanation
- **Revised abstract** to more accurately reflect the focus on G-quadruplex structures while appropriately contextualizing the broader applicability of BIVID-MaP

Below, we provide detailed point-by-point responses to each comment.

1. The rate of reactivity of the BVQ probes is really slow (up to 28 hours!). Would this approach ever be applicability in a cellular context? While investigating in vitro transcribed RNAs is useful, many complexities of the cell may preclude formation of a structure, especially RNA G-quadruplexes which have been demonstrated to be fleeting, especially in 5' UTRs that are unwound by the ribosome. Discussion of this aspect should be added, as well as the potential biological relevance of the discovered motifs. There is major concern that many of these findings will not translate to cells, as has been observed with other RNA G-quadruplexes (e.g., in the NRAS transcript).

Response 1):

We thank the reviewer for this critical comment regarding reaction kinetics and the biological relevance of our findings in cellular contexts.

We fully appreciate the concern about the slow reaction rate (up to 28 hours) and its implications for cellular applicability. To address this, **we have developed a new activation method using methyl vinyl ketone (MVK)** that dramatically reduces the reaction time to 10-30 minutes (**Extended Data Fig. 9a, b**). **Importantly, we successfully demonstrated that BIVID-MaP detects berberine-G4 interactions in cell lysate with only 10 minutes reaction time (Extended Data Fig. 9c–e)**. This is comparable to the reaction times typically employed in conventional mutational profiling and crosslinking-based binding detection methods (Zeller, M. J. et al., 2022, PNAS, Fang, L. et al., 2023, Nat. Chem.).

Importantly, the presence of glutathione in cellular environments was initially a concern, as it could quench VQ reactivity. However, **MVK serves a dual function: it both accelerates the reaction and blocks competing reactions with cellular thiols**, enabling efficient labeling even in complex biological environments.

The dramatically shortened reaction time (10-30 min) of our new method may enable capture of transiently formed G-quadruplex structures that would be missed with longer reaction times, as the rapid labeling can occur before these structures are disrupted.

While our in vitro findings provide valuable insights into SNV-dependent RNA structural changes and small-molecule binding, we now clearly state that cellular validation will be valuable to fully characterize the biological and therapeutic relevance of the discovered motifs.

Changes made to the manuscript:

- Added new data showing rapid reaction method with MVK and cell lysate validation (**Extended Data Fig. 9**)
- Added new text in the Discussion sections, 8th paragraph: *“Considering the broader application of BIVID-MaP, fast-reacting modification chemistry is well-suited to the dynamic and unstable RNA structural environment in cells^{45–47}, where G4 structures, in particular, are often unfolded under complex cellular contexts⁴⁸. Importantly, we demonstrated that pre-activation of the VQ-precursor by a 4-hour incubation with methyl vinyl ketone (MVK) enables rapid modification, even when the reaction time is shortened to 10 minutes (Extended Data Fig. 9a, b). This pre-activation allowed us to detect berberine-G4 interactions in HeLa cell lysates with sensitivity similar to that observed in the buffer, confirming that a 10-minute incubation is sufficient for detecting berberine binding (Extended Data Fig. 9c–e). A possible concern is that glutathione in the cellular environment might react with VQ, which could suppress its modification efficiency. In our reaction system, MVK not only accelerates the reaction but also minimizes undesired reactions with intracellular thiols by acting as a trapping reagent. This enables efficient modification even under conditions that mimic the complex cellular environment, thus suggesting the feasibility of extending BIVID-MaP to intracellular applications, although further optimization of*

the VQ-conjugated compounds, including improvements in membrane permeability, is still needed.”

2. Using the TGIRT for deletions rather than mutations is a very interesting finding, yet biochemical details regarding why this enzyme can do this are missing. Additional details should be provided regarding this unique reactivity, particularly since the method relies heavily on this previously unknown reaction.

Response 2):

We thank the reviewer for highlighting the importance of understanding the biochemical basis for TGIRT's deletion-generating mechanism.

To investigate the molecular mechanism by which VQ modification generates deletions during reverse transcription, **we additionally performed molecular modeling and molecular dynamics (MD) simulations** using VQ-modified uracil as the template base (**new Supplementary Fig. 1**). Our simulations show that the VQ moiety stacks with the terminal cDNA base, occupying the incoming nucleotide position, consistent with prior studies on DNA polymerase-induced frameshift deletions, suggesting that planar modifications promote deletions by offering a larger stacking surface than canonical bases (Zhang H. et al, 2009, *J Biol Chem*). This observed stacking persists in the complex structure with TGIRT, which also explains why TGIRT causes nucleotide deletion in VQ-modified bases.

This distinct behavior makes **TGIRT uniquely suitable for BIVID-MaP**, as the deletion signatures it generates provide specific, positional information about RNA-small molecule binding sites.

Changes made to the manuscript:

- Added molecular dynamics simulation data showing VQ stacking at the template position (**Supplementary Fig. 1**)
- Added detailed explanation of TGIRT's hypothesized deletion-generating mechanism in the Results sections, the 1st to 3rd paragraph: *“Previous research on frameshift deletions induced by DNA polymerase indicates that planar modifications, such as 1, N2-etheno-dG, induce deletions by providing a larger stacking surface than canonical bases⁹. The π - π stacking interactions of the modified base with the terminal base of the elongating strand and the incoming dNTP are considered crucial for this process. Based on this mechanism, we hypothesized that a similar modification on RNA would induce RT deletions”, “We previously reported a modifier comprising an RNA-binding small molecule (Rbs) conjugated to a dimethylamino-protected vinyl-quinazolinone, termed VQ (NMe₂)^{10, 11}. This Rbs-VQ (NMe₂) can be activated to the vinyl VQ form, enabling covalent bonding with bases, especially uridine bases (U), near a target RNA structure. The N3 position of U is the primary modification site. We investigated whether this VQ-modified U could induce deletions near the Rbs binding sites.”, “Our molecular modeling and MD simulation analyses of an RNA/cDNA complex containing a VQ-modified base suggest that the VQ moiety stacks with the terminal base of the growing cDNA strand (at the -1 position), thereby occupying the incoming nucleotide binding position (**Supplementary Fig. 1a, b**). This stacking persisted in the complex with Thermostable Group II Intron Reverse Transcriptase (TGIRT)¹², a commonly employed RT enzyme in conventional mutation profiling⁸ (**Supplementary Fig. 1c**). We also observed a potential pocket for accommodating the Rbs and linker regions of the VQ modifier. To develop BIVID-MaP for detecting variant-specific interactions of small molecules, we first assessed how VQ-mediated covalent modification induces the three main types of RT mutations (i.e., substitution, deletion, and insertion)”*

3. Discussion of how affinity affects labeling should be provided. Based on the slow kinetics of the reaction, is there an affinity threshold that must be met?

Response 3):

We thank the reviewer for this important point. To address the relationship between binding affinity and labeling efficiency, we analyzed the K_d values of RNA-compound pairs that showed detectable signals in BIVID-MaP.

Detected pairs showed the following affinities:

- Loop GAAGGAAGG- SMN-C2: 4.6 μM (competitive fluorescence titration, Tang et al., NAR, 2021)
- G4 TERRA-berberine: 0.47 μM (Yutong Chen et al., Bioconjugate Chem., 2022)
- G4 TERRA-CMA (Acridine): 1.0 μM (Yutong Chen et al., Bioconjugate Chem., 2022)

A key point is that the threshold for detectable binding affinity in BIVID-MaP depends on the compound concentration used. The following preliminary experimental results support this point.

- **Loop GAAGGAAGG - SMN-C2:** SMN-C2-VQ barely modified the Loop GAAGGAAGG at 16 μM , but increasing the probe concentration to 40 μM resulted in a clear modification signal observed in gel analysis (**as shown in the Figure below**). This indicates that increasing the compound concentration enhances labeling efficiency and expands the range of binding affinities detectable by BIVID-MaP.

Figure | Gel shift assay showing modifications to RNA structure motifs by SMN-C2 (NMe₂)

The gel image shows the concentration-dependent modification of SMNV2-VQ (NMe₂) on Loop GAAGGAAGG (1 μM) after 28 h in buffer (20 mM phosphate pH 7.0, 20 mM NaCl, 80 mM KCl). From bottom to top, the gel shift bands show, unmodified RNA, a 1:1 covalent complex (one modifier per RNA) and 1:2 covalent complex (two modifiers per RNA).

As the reviewer pointed out, the relationship between binding affinity and labeling efficiency is an important guideline for BIVID-MaP users. We have clarified these points in the Discussion section.

Changes made to the manuscript:

- Added the new text in Discussion section, 7th paragraph: *“Our research demonstrated that BIVID-MaP can detect RNA-small molecule interactions with K_d values ranging from sub-micromolar to low micromolar ($K_d = 0.47 \mu\text{M}$ for the berberine-G4 TERRA interaction¹¹ and $K_d = 4 \mu\text{M}$ for the SMN-C2-Loop GAAGGAAGG interaction²¹). The range of detectable binding affinities depends on the compound concentration used in the experiment and the affinity of each RNA-small molecule pair. Increasing the compound concentration improves modification efficiency, thereby extending the range of affinities that can be detected by BIVID-MaP towards weaker interactions.”*

4. In several of the RNA gels, it appears that the RNA is being degraded over time (e.g., SMN-C2 in Figure 1C and CD44 in Figure 2E). Are the RNAs being degraded or can the RNA no longer be detected? Details regarding visualization of the RNA gels is missing in the methods. Is a dye being used that can no longer bind following probe labeling?

Response 4):

We thank the reviewer for this careful observation and for identifying the missing methodological details. To address these concerns, we have thoroughly re-examined our gel data.

We did not observe a pronounced, modification time-dependent increase in smearing or shorter RNA fragments, suggesting that extensive RNA degradation is unlikely. Instead, we interpret the reduction in band intensity as being primarily due to a loss of fluorescence upon modification. Particularly obvious in SMN-C2, the band intensity of highly reactive RNA is markedly diminished compared to that of low-reactivity RNA (as shown in the Figure below). Because we visualized the gels using 5'-FAM-labeled RNA, extensive modification by Rbs-VQ is likely to perturb the FAM fluorophore, resulting in reduced apparent band intensity.

Figure | Gel shift assay with SMN-C2-VQ (NMe₂) or Berberine-VQ-N₃ (NMe₂)

(a–b) Gel shift assay measuring the modification rate of SMN-C2-VQ (NMe₂) **(a)** or Berberine-VQ-N₃ (NMe₂) **(b)**. In each target small molecule, Loop GAAGGAAGG or CD44 G4 WT was used as a high reactive RNA while Loop G-bulge or CD44 G4 Mut was used as a low reactive RNA. From bottom to top, the gel shift bands show unmodified RNA, a 1:1 covalent complex (one modifier per RNA) and a 1:2 covalent complex (two modifiers per RNA).

We fully agree with the reviewer that our explanation of RNA gel visualization, including the use of FAM, was insufficient. We have now clarified the explanation regarding visualization in the Methods Section.

Additionally, we applied another strategy for gel visualization of G4 HIV-1 LTR, the longest in our targets, which is also not demonstrated in the previous manuscript. **We have now added a more detailed description of these procedures in the Methods section.**

Changes made to the manuscript:

- Additional rows have been incorporated into **Supplementary Data 4** to specify which fluorescence detection system is used for each sequence.
- Added new text in Figure legends (**Fig. 1d**): *"For visualization, SYBR Gold staining was used for the G4 HIV-1 LTR, and 5'-FAM labeling was used for the Loop GAAGGAAGG "*
- Added new text in Figure legends (**Fig. 2e, Supplementary Fig. 4**): *"5'-FAM labeled RNA was used for gel visualization."*
- Added the more detailed explanation about gel visualization in Methods section entitled **Confirmation of modification by gel shift assay (Except G4 HIV-1 LTR)**: *"For gel visualization, RNAs were labeled with 5'-FAM (fluorescein) at the 5' terminus."*

- Added the new section showing the detailed protocol for long RNA in Methods section entitled **Confirmation of modification by gel shift assay for G4 HIV-1 LTR**: *“For the G4 HIV-1 LTR RNA (64 nt), the longest sequence in which we performed gel shift assays, adduct size was increased by post reaction to facilitate detection of gel mobility shifts. Non-FAM-labeled RNA was purchased from Agilent. RNA design, folding, and primary modification were performed as described above. For modification, an azide-attached Rbs-VQ precursor was used. The resulting azide-modified RNA (10 μL) was subjected to a click reaction by adding 2 μL of 50 mM DBCO-PEG4-amine and incubating at 37 °C for 3 h. Next, a 12 μL aliquot of the solution was mixed with 12 μL of loading buffer. A 6 μL sample of this mixture was then loaded onto a 16% denaturing polyacrylamide gel (containing 20% formamide in TBE buffer) and subjected to electrophoresis at 500 V for 90–100 minutes at room temperature. After electrophoresis, RNA was stained with SYBR Gold and imaged using a ChemiDoc Touch MP system (Bio-Rad). Band intensities were quantified using Image Lab software (Bio-Rad). Detailed RNA sequences are provided in **Supplementary Data 4.**”*

5. Again, regarding the gels, it is surprising that a detectable gel shift is observed considering the size of the RNAs. The authors should provide evidence that labeling stoichiometry is 1:1 (some seem like additional bands are formed over time). Along similar lines, probe to RNA equivalents should be added to the figure legend.

Response 5:

We thank the reviewer for this important comment.

As the reviewer noted, the larger the RNA, the more difficult it becomes to detect gel shifts. We employed click chemistry to increase the size of the adduct, specifically for the G4 HIV-1 LTR (64 nt), to facilitate detection of the gel mobility shift. However, as stated in Response 4, this explanation was totally insufficient in the previous manuscript, so **we have added further details in the Method section.**

To address the concern about labeling stoichiometry and the observed gel shifts, we performed additional characterization experiments, targeting (UGGU)₆, which exhibited the clearest additional band when modified by Berberine-VQ-N₃ (NMe₂) (**Supplementary Fig. 4**). MALDI-TOF mass spectrometry revealed the presence of RNAs corresponding to each one or two alkylation events, showing that the additional band observed in gel-shift assay represents 1:2 stoichiometry (**New Supplementary Fig. 18**). **We have clearly indicated the labeling stoichiometry of multiple bands in the Results section and in the legends of all figures including gel images.**

Changes made to the manuscript:

- Added new MALDI-TOF mass spectrometry data confirming the labeling stoichiometry for representative RNA-probe pairs (1:1 for single-modified RNAs; 1:2 for dual-modified RNAs) (**Supplementary Fig. 18**).
- Added the new text in the figure legend to clarify the labeling stoichiometry for each gel band in **Fig. 1d, Fig. 2e, Extended Data Fig. 9b and Supplementary Fig. 4**: *“From bottom to top, the gel shift bands show, unmodified RNA, a 1:1 covalent complex (one modifier per RNA) and 1:2 covalent complex (two modifiers per RNA) (See Supplementary Note).”*
- Added the new text in Supplementary Note entitled **MALDI-TOF analysis indicates that the multiple gel-shift bands originate from 1:1 and 1:2 Rbs-VQ:RNA labeling stoichiometries**: *“As multiple band shifts were observed in the gel shift assay (Fig. 1d, Fig. 2e, Supplementary Fig. 4), we verified their stoichiometric origin. MALDI-TOF analysis of model G4 RNA (UGGU)₆ revealed that both single and double alkylation events per RNA molecule occurred (Supplementary Fig. 18). This result suggests that the formation of multiple adducts on a single*

RNA molecule increases the probability of at least one RT deletion per molecule, thereby potentially amplifying the RT-deletion effectively.”

- Added the new section showing the detailed protocol for long RNA in Methods section entitled **Confirmation of modification by gel shift assay for G4 HIV-1 LTR**: *“For the G4 HIV-1 LTR RNA (64 nt), the longest sequence in which we performed gel shift assays, adduct size was increased by post reaction to facilitate detection of gel mobility shifts. Non-FAM-labeled RNA was purchased from Agilent. RNA design, folding, and primary modification were performed as described above. For modification, an azide-attached Rbs-VQ precursor was used. The resulting azide-modified RNA (10 µL) was subjected to a click reaction by adding 2 µL of 50 mM DBCO-PEG4-amine and incubating at 37 °C for 3 h. Next, a 12 µL aliquot of the solution was mixed with 12 µL of loading buffer. A 6 µL sample of this mixture was then loaded onto a 16% denaturing polyacrylamide gel (containing 20% formamide in TBE buffer) and subjected to electrophoresis at 500 V for 90–100 minutes at room temperature. After electrophoresis, RNA was stained with SYBR Gold and imaged using a ChemiDoc Touch MP system (Bio-Rad). Band intensities were quantified using Image Lab software (Bio-Rad). Detailed RNA sequences are provided in **Supplementary Data 4.**”*

6. The title and abstract imply much greater impact with respect to SNVs than is described in the text. From my reading, the manuscript is focused almost solely on G-quadruplexes, and as such, the title and abstract should be rewritten to be more accurate. The manuscript largely describes a method (BIVID-MaP) that can be used to probe for G-quadruplex-containing RNAs; however, questions remain regarding the method due to the weak characterization of the TGIRT reactivity as described above in point 2.

Response 6):

We thank the reviewer for this important comment regarding the scope and positioning of our manuscript. We have carefully reconsidered our presentation to more accurately reflect the breadth and focus of BIVID-MaP.

We acknowledge that our study has primarily focused on variant-specific binding detection in RNA G-quadruplex structures. To reflect this more accurately, **we have revised the Abstract and Introduction to specify the focus on G-quadruplex structures, adjusting the scope of our claims more appropriately.**

Clarification on the scope of BIVID-MaP: While BIVID-MaP is designed as a general platform for detecting small molecule-RNA interactions, we acknowledge that our SNV validation has focused primarily on G-quadruplex structures. However, we have demonstrated that the method itself is not limited to G-quadruplexes, which was not conveyed previously:

- SMN-C2 interacts with non-G4 RNA structures (GAA repeat loop) (**Supplementary Fig. 2**): We added additional experimental data to show our method detects hairpin loop specific binding of SMN-C2.

Regarding the characterization of the TGIRT reactivity raised in point 2, we have addressed this with additional MD simulation and experimental characterization (see Response 2).

Changes made to the manuscript:

- Changed the text in the Abstract to clarify our discovery related to variant-specific interactions was limited to G4 structures: *“Using BIVID-MaP, we uncovered numerous variant-specific interactions between a G-quadruplex (G4)-binding small molecule and RNAs harboring*

*single-nucleotide variants. Several cancer-associated somatic mutations significantly influence the binding intensity of a small molecule by affecting target **G4 structures***

- Changed the text in the Introduction to clarify that our discovery of variant-specific RNA structure-small molecule interactions was limited to G4 structures: *“Using this approach, BIVID-MaP discovered numerous cancer-associated somatic mutation-specific interactions between 5' UTRs and a **G-quadruplex (G4)-binding small molecule.**”*
- Added the new text and data to clarify that the BIVID-MaP itself is adaptable to other RNA structures (**Supplementary Fig. 2**) in Results section, 5th paragraph: *“Furthermore, to demonstrate whether our method can detect binding to different types of RNA structures, we tested SMN-C2, which is an analog of an FDA-approved RNA-targeting drug for spinal muscular atrophy²⁰. SMN-C2-VQ (NMe₂) treatment resulted in significant RT deletions in the target hairpin structure in Loop GAAGGAAGG²¹ (**Fig. 1c-f**). Notably, the significant RT deletions were diminished when target purine-rich sequences were in the stem structure, meaning that our method recognizes the selective binding to the loop structure of SMN-C2 (**Supplementary Fig. 2**). These results demonstrate that the VQ-conjugated RNA-binding moiety can be replaced with other RNA-binding compounds.”*

Reviewer 1

The authors have provided appropriate and satisfactory responses to the reviewer's comments, including a comparison with competing technologies. This reviewer considers the manuscript suitable for publication in Nature Communications.

Response:

We sincerely thank the reviewer for the positive feedback and for supporting publication of our manuscript.

Reviewer 2

The authors have done an excellent job in addressing all my comments. I therefore recommend the publication of the manuscript in Nature Communications.

Major comments:

1. OK.
2. OK.
3. OK.
4. OK.
5. OK.
6. OK.
7. OK.
8. OK.

Minor comments:

1-4. All OK.

Response:

We sincerely thank the reviewer for the supportive evaluation and the recommendation for publication.

Reviewer 3

In this revised manuscript, the authors have largely addressed the comments from the previous critiques. In doing so, however, additional points to be addressed have been identified that will be outlined below:

Response:

We sincerely thank the reviewer for the positive feedback and insightful comments, which have helped us further improve and strengthen the manuscript.

1. Like the abstract, the title should also be updated to include G-quadruplex specificity.

Response:

We thank the reviewer for the important suggestion to explicitly reflect the G-quadruplex focus of this study in the title, consistent with the revised Abstract.

Following the reviewer's suggestion, we have modified the title to specifically emphasize G-quadruplex structures, which more accurately reflects the scope of our claims.

Change made to manuscript:

- The title was updated to: *"Systematic identification of variant-specific RNA structure-small molecule interactions, exemplified by RNA G-quadruplex structures"*

2. How does MVK accelerate the reaction? Additional details should be provided or this should be removed and the subject of a future manuscript.

Response:

We thank the reviewer for the important comment regarding how MVK accelerates the reaction.

MVK acts as a trapping reagent for the eliminated secondary amine during VQ generation. By irreversibly capturing this leaving group, MVK shifts the equilibrium toward the active VQ species and effectively suppresses the reverse reaction, thereby accelerating the overall labeling process.

Following the reviewer's suggestion, we have incorporated a more detailed explanation regarding how MVK accelerates the reaction in the Discussion section.

Change made to manuscript:

- We have added the following text in the Discussion section, 8th paragraph: *"During this pre-incubation, MVK facilitates VQ generation through the irreversible capture of the eliminated secondary amine. This process shifts the chemical equilibrium toward the formation of the active VQ species and prevents the reverse reaction, thereby bypassing what was previously the rate-limiting elimination step."*

3. I find it concerning that SHAPE-MaP and DMS-MaP were not able to detect any RNA-small molecule interactions identified using BIVID-MaP. Have the authors identified other RNA-ligand systems that worked for all 3 methods. More comments regarding this lack of overlapping detection should be included in the discussion.

Response:

We thank the reviewer for the important comment regarding the limited overlap in detection across all three methods.

In our validation experiments, no RNA-ligand interactions were consistently detected by all three methods. This can be explained by the following three reasons:

- Ligand binding does not always alter the modification patterns in SHAPE-MaP or DMS-MaPseq.** DMS-MaPseq primarily detects the mutation signals at A and C bases, whereas SHAPE reagents modify the ribose 2'-OH. As ligand binding does not necessarily occur at these specific sites, interactions that do not appreciably alter these reactivity profiles can be missed using these approaches.
- G4 structures often exhibited elevated background mutation rates in SHAPE-MaP (as shown in the Figure below).** This higher background makes it more challenging to detect modifier- or ligand-dependent mutation signals and **may therefore explain why most interactions were not detected by SHAPE-MaP in our case.**

Figure | G4 structures exhibit elevated background mutation rates in SHAPE-MaP

RT-mutation (%) profiles in SHAPE-MaP or DMS-MaPseq in the absence of the target small molecule. Blue and red lines indicate conditions with the chemical modifier, whereas gray lines indicate no-modifier controls. Panels (a) and (b) show sequences forming G4 structures, and panel (c) shows a non-G4 sequence. Shaded bands indicate the range between the two replicates. The G-tract-forming guanines are highlighted by underlining. In DMS-MaPseq, modifier-dependent increases in RT-mutation are clearly observed at adenine and cytosine nucleotides. In contrast, in SHAPE-MaP, G4 structure forming sequences exhibit elevated background mutation rates, making modifier-dependent increases difficult to detect.

- Nucleotide selectivity differs across methods.** BIVID-MaP probes U-dependent labeling, whereas DMS-MaPseq primarily detects mutation signals at A and C bases. As a result, U-free RNA (G4 pre-mir-6850 in **Extended Data Fig. 5**) is not detected by BIVID-MaP, while A/C-free RNA (G4 (UGGU)₆ in **Extended Data Fig. 5**) is not detected by DMS-MaPseq, thereby reducing overlap across methods.

As the reviewer mentioned, these explanations were not sufficiently provided in the previous manuscript. **We have added more detailed explanations to the Results section and Supplementary Note section.**

Change made to manuscript:

- Added the text in the Results section, 8th paragraph specifying the limited overlap in detectable interactions across the three methods: *“To evaluate the performance of BIVID-MaP relative to other mutational profiling methods, we conducted SHAPE-MaP^{7, 22, 23} and DMS-MaPseq⁸ in the presence and absence of RNA-binding ligands to compare mutation rates (Extended Data Fig. 5a). SHAPE-MaP failed to detect RNA-small molecule interactions identified by BIVID-MaP, as no significant differential RT-mutations were observed in positive controls compared to negative controls (Extended Data Fig. 5b, Supplementary Figs. 5–7). DMS-MaPseq missed some berberine-G4 structure interactions (G4 pre-mir-1229, G4 BCL2, G4 (UGGU)6), which are clearly detected by BIVID-MaP (Extended Data Fig. 5b, Supplementary Fig. 5). Furthermore, DMS-MaPseq could not detect SMN-C2 binding to GA-rich hairpin loops (Supplementary Fig. 7c, d). These method-dependent outcomes may reflect mechanistic differences between mutational profiling methods (Supplementary Note). Overall, these results demonstrate that BIVID-MaP provides superior sensitivity and specificity for detecting RNA-small molecule interactions in our validation system.”*
- Added the detailed explanation in the Supplementary Note section entitled **“The mechanistic differences between mutational profiling approaches can impact RNA-small molecule interaction detection”**: *The limited detection capability of other mutational profiling approaches can be partially explained by their modification and detection preferences. DMS-MaPseq primarily detects mutation signals arising from DMS modifications at A and C bases. SHAPE-MaP, in contrast, modifies the ribose 2'-OH^{7, 8}. However, direct ligand binding does not always occur at these specific sites or produce detectable changes in their reactivity. This may explain why these approaches do not consistently detect RNA–small molecule interactions, and why interactions that do not appreciably change DMS or SHAPE reactivity profiles are missed. Second, baseline background and reverse transcription performance can further limit detectability for highly structured RNAs. In our SHAPE-MaP measurements, G4 structure forming sequences exhibited elevated baseline mutation rates even under no-modifier/no-ligand control conditions. This higher background reduces the signal-to-noise ratio and can mask modifier- or ligand-dependent changes, making interaction signals more difficult to detect in SHAPE-MaP compared with other methods in our validation system. Finally, nucleotide selectivity varies between methods, which can restrict the generation of signals for specific target sequences. BIVID-MaP uses U-dependent labelling, whereas DMS-MaPseq detects mutation signals at A and C bases. Consequently, U-free RNA (G4 pre-mir-6850 in Extended Data Fig. 5) is not detected by BIVID-MaP, and A/C-free RNA (G4 (UGGU)6 in Extended Data Fig. 5) is not detected by DMS-MaPseq. This reduces the overlap between the two methods. Together, these differences in modification preferences, reverse transcription performance, and nucleotide selectivity help explain the limited overlap among methods.*